



# Cushion bog plant community responses to passive warming in southern Patagonia

Verónica Pancotto[1,2], David Holl[3], Julio Escobar[1], María Florencia Castagnani[1], and Lars Kutzbach[3]

[1]Centro Austral de Investigaciones Científicas (CADIC-CONICET), Ushuaia, Argentina
[2]Universidad de Tierra del Fuego (ICPA-UNTDF), Ushuaia, Argentina
[3]Institute of Soil Science, Center for Earth System Research and Sustainability (CEN), Universität Hamburg, Hamburg, Germany

**Correspondence:** David Holl (david.holl@uni-hamburg.de)

**Abstract.** Vascular plant-dominated cushion bogs, which are exclusive to the southern hemisphere, are highly productive and constitute large sinks for atmospheric carbon dioxide compared to their moss-dominated counterparts around the globe. In this study, we experimentally investigated how a cushion bog plant community responded to elevated surface temperature conditions as they are predicted to occur in a future climate. We conducted the study in a cushion bog dominated by *Astelia*

*pumila* on Tierra del Fuego, Argentina. We installed a year-round passive warming experiment using semicircular plastic walls that raised average near-surface air temperatures between 0.4 °C and 0.7 °C (n=3). We focused on characterizing differences in morphological cushion plant traits and in carbon dioxide exchange dynamics using chamber gas flux measurements. We used a mechanistic modeling approach to quantify physiological plant traits and to partition the net carbon dioxide flux into its two components photosynthesis and total ecosystem respiration. We found that *A. pumila* reduced its photosynthetic activity under

elevated temperatures. At the same time, we observed enhanced respiration which we attribute, due to the limited effect of our passive warming on soil temperatures, largely to an increase in autotrophic respiration. Passively warmed *A. pumila* cushions sequestered between 55 % and 85 % less carbon dioxide than untreated control cushions over the main growing season. These results suggest that future warming could decrease the carbon sink function of austral cushion bogs.

**1   Introduction**

Peatlands are an important component of the global carbon cycle due to the long-term accumulation of organic matter in peat soils (Gorham, 1991; Parish et al., 2008; Alexandrov et al., 2020). Carbon (C) accumulated in peatland soils over the past millenia due to the greater carbon uptake via photosynthesis (gross primary production, GPP) compared to the carbon lost through ecosystem respiration, methane emissions and waterborne export. Peatlands represent a C pool (550 Gt; Yu et al.,

2010) of global importance. In a warmer climate, organic matter decomposition in peatlands could be enhanced (Broder et al., 2012, 2015). As a result, increased amounts of carbon dioxide could be released to the atmosphere and act as a positive



feedback on global warming. Peatland $CO_2$ dynamics are mainly controlled by radiation, temperature, soil water content and plant community composition. Ecosystem respiration generally responds positively to increased temperatures (Gallego-Sala et al., 2018) and negatively to oxygen-depleted soil conditions (Wilson et al., 2016). Water-saturation in soils typically leads

to low oxygen availability and thus to low heterotrophic respiration. However, if vascular plants are present, they can transport oxygen into water-saturated soil layers and create oxic zones close to their roots, where respiration can be enhanced.

We conducted our study in a southern hemisphere bog on Tierra del Fuego, Argentina. In contrast to northern hemisphere bogs, our study site is dominated not by mosses but by vascular peat-forming cushion plants, which exclusively exist on the southern hemisphere. These plants are characterized by a dense root and rhizome system and a large belowground biomass.

Oxygen-transport has been shown to be efficient in these systems (Fritz et al., 2011; Münchberger et al., 2019) leading to close-to-zero methane emissions from cushion bogs. Furthermore, cushion bogs currently act as strong carbon dioxide net sinks as recently reported by Holl et al. (2019) when compared to moss-dominated bogs, which are typical for the northern hemisphere. Apart from the mentioned publications, to date, little is known about cushion bog carbon exchange dynamics as well as the possible alterations of these dynamics in a changing climate. Air temperature in the southern hemisphere is

projected to increase by 0.4 °C – 0.6 °C per decade in the near future (2025-2049) under the SSP1-2.6 and SSP2-4.5 scenarios (Fan et al., 2020). Soil moisture is predicted to diminish due to precipitation decrease and temperature increase leading to higher soil evapotranspiration.

We conducted a field experiment to determine how cushion-forming plants respond to experimental warming. In particular, we investigated the effect of experimental warming on morphological and physiological cushion plant traits. We manipulated

the temperature conditions passively with open side chambers (OSCs), which we set up as u-shaped plastic walls with the walls mounted perpendicular to the soil surface. An OSC acts as a "solar energy trap" (Marion et al., 1997) primarily by reducing radiative heat loss (Aronson and McNulty, 2009). Chamber design and local conditions can introduce a variety of secondary effects (discussed in section 4) that can additionally impact the energy balance and surface temperature of a treated plot. We chose this method as it has successfully been used in ecosystem warming studies at high latitudes (Marion et al., 1997; Aronson

and McNulty, 2009), is cost-effective and appropriate for remote sites with limited power supply.

## 2   Material and Methods

### 2.1   Study site

The study was carried out in a cushion plant-dominated bog (cushion bog) located on Tierra del Fuego, Argentina (54.973° S, 66.734° W). Evergreen forests dominated by *Nothofagus betuloides* and bogs are typical features of the windy coastal areas in

southeast Tierra del Fuego (Ponce and Fernández, 2014; Grootjans et al., 2010; Kleinebecker et al., 2007). The microrelief at the cushion bog is flat and consists of a patterned surface with lawns of the cushion-forming plant *Astelia pumila* and around 50 cm deep pools of various sizes (between around 0.5 m$^2$ and 10 m$^2$, cf. Münchberger et al. (2019)). Small patches dominated by *Sphagnum magellanicum* or cushion-forming *Donatia fascicularis* are embedded in these *Astelia lawns*. *A. pumila* establishes a dense, aerenchymatic root and rhizome system reaching up to 2 m below the soil surface (Fritz et al., 2011; Münchberger





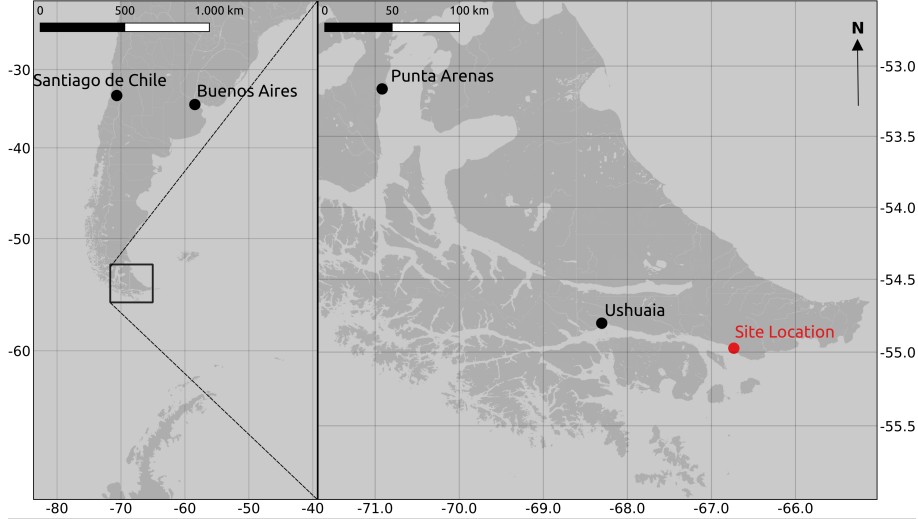

**Figure 1.** Location of the study site, a cushion bog at the northern coast of the Beagle Channel, close to Punta Moat on Isla Grande de Tierra del Fuego, Argentina.

et al., 2019). Other species such as *Tetroncium magellanicum, Gaultheria antarctica, Caltha dioniflora, Drosera uniflora, Empetrum rubrum, Nanodea muscosa, Pernettya pumila*, and *Myrteola nummularia* are frequently present, though with very low cover (totaled up less than about 20 % areal cover, see Table B2). The climate in the southernmost part of the Fuegian Archipelago is highly oceanic, cold-humid, with mild winters, and typical winds from the southwest, which are most intense in spring (Santana et al., 2006; van Bellen et al., 2016). Meteorological variables (air and soil temperature, relative humidity,

precipitation, wind direction, wind velocity and photosynthetically active radiation) were recorded at 1 minute intervals and averaged over 30 minutes on a data logger (CR3000; Campbell Scientific, UK) at the nearby eddy covariance system, installed in February 2016 (cf. Holl et al., 2019). Between 1 September 2016 and 31 August 2019, we measured 530 mm to 790 mm annual precipitation and average annual air temperatures between 5.9 °C and 6.6 °C (see Table B1). Long-term meteorological observations do not exist for the exact location of our site. For the city of Ushuaia (around 100 km west from our site), the

long-term average annual precipitation sum is reported to amount to 530 mm to 574 mm (Iturraspe, 2012; Tuhkanen, 1992). Also for Ushuaia, Iturraspe (2012) reports a mean annual temperature of 5.5 °C. According to the eddy covariance data from our site, the most frequent wind directions (sorted by abundance in descending order) were north-northwest, west-southwest and west-northwest between 25 January 2016 and 17 May 2018 (see supplementary Figures S33 and S34).

## 2.2 Setup of warming experiment

In October 2014, we installed twenty plots; ten of which were randomly assigned to receive warming treatment (treatment plots), where air and soil temperatures were passively increased by placing open-side chambers (OSCs) over the vegetation similar to the approach of Marion et al. (1997). The OSC, made of transparent polycarbonate in the shape of a semicircle open



to the north, is 50 cm high and 1.30 m in diameter (Figure 2). We measured the transmittance of a small section of polycarbonate in the visible region of the daylight spectrum with a spectrophotometer (UV-1203, Shimadzu Corporation, Japan), showing a

high transmittance, ranging between 80 % and 85 %. Air temperature 1 cm above the canopy and soil temperature 10 cm below the surface were recorded hourly inside and outside the treatment plots in three replicates using HOBO U12 data loggers (Onset Computer Corporation, USA) between 01 January 2018 and 18 January 2019. The air probes were covered by shade caps to avoid direct sunlight while allowing free airflow around the probe. To evaluate if the treatment actually did affect temperatures, we calculated the differences between the treatment and the related control temperature measurements at each time step and

calculated the mean difference. We then applied a randomization test (Edgington and Onghena, 2007) by randomly assigning each measurement to the treatment or control group and again calculating the mean difference. We repeated this step 10000 times. Next, we compared the distribution of the 10000 mean differences calculated with random treatment assignments to the actual mean difference between treatment and control using a one-sample t-test. We regard the case when the actual mean is not likely (p < 0.01) to come from the same distribution as the randomized mean differences as evidence for a significant

treatment effect.

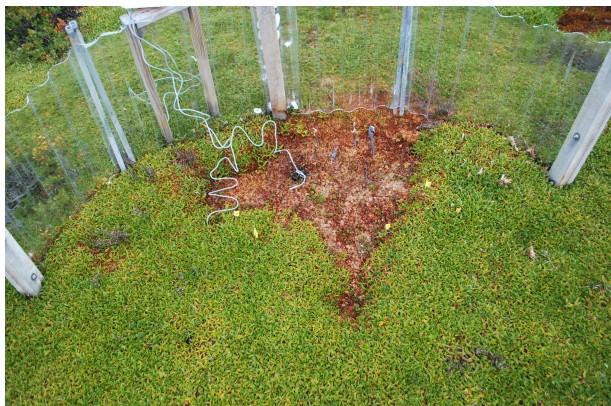

**Figure 2.** Experimental setup. Open side chamber (OSC), open to the north, installed in an *Astelia pumila* (light green color) cushion with a patch of *Sphagnum magellanicum* (reddish-brownish color)

## 2.3 Leaf properties

During two growing seasons (2015/2016 and 2017/2018), we conducted length measurements of *Astelia pumila* leaves and counted the number of total, green, and senescent leaves (see supplementary Tables S3 to S5). We selected 10 individuals per plot, labeled and measured the length of a young, fully expanded leaf at the beginning of the growing season using a

digital caliper. We marked the measured leaves and determined the lengths of the same leaves at the middle and at the end of the growing season. From the treatment as well as the control plots, we recorded the lengths of 100 individual *Astelia* leaves for each group during the growing seasons of 2015/2016 and 2017/2018 at three (January, February, April 2016) and two (September 2017, March 2018) points in time, respectively. By subtracting the lengths of single leaves, which we tracked





individually throughout two growing seasons, late in the season from the respective lengths at the beginning of the season, we

estimated and compared average leaf growth at the treatment and control plots.

We estimated the area of 86 leaves by analyzing vegetation pictures taken at the different plots during 12 measurement days between 15 January 2016 and 04 March 2016 (see supplementary Table S2). We calibrated and processed the images using the software ImageJ (Rueden et al., 2017). We divided the area estimates into two groups referring to midsummer and late summer and compared the respective treatment and control means using a Mann-Whitney U-test of the SciPy Python library (Virtanen

et al., 2020). Additionally, we determined the specific leaf area (SLA, cm2/g), leaf water content (LWC, %), and leaf dry mass content (LDMC, mg/g). We estimated dry mass after drying the leaves in an oven at 65 °C for 72 hours or to constant weight. We estimated LWC and LDMC following Pérez-Harguindeguy et al. (2016).

On six days between January 2016 and September 2017, we sampled fully expanded *Astelia* leaves and extracted their photosynthetic pigments (chlorophyll and carotenoids, see supplementary Table S6) using dimethyl sulfoxide solution (DMSO).

We collected six leaves per plot and cut two 7.5 mm$^2$ large discs from each leaf, resulting in 12 discs per plot. We macerated the discs in 5 ml of DMSO and heated them to 70 °C for 45 minutes (Barnes et al., 1992; Wellburn, 1994). We measured the absorbances at wavelengths of 665 nm, 649 nm and 480 nm with a spectrophotometer (UV-1203, Shimadzu Corporation, Japan). We followed Wellburn (1994) to calculate chlorophyll *a*, chlorophyll *b* and carotene contents.

To investigate the treatment effect on the number of plants per area, we counted the density of plant individuals in about

10 cm by 10 cm large areas within 10 treatment and 10 control plots (see supplementary Table S1) in April 2016.

## 2.4 Chamber soil–atmosphere CO$_2$ flux measurements

We used a closed static chamber technique (Livingston and Hutchinson, 1995) to determine CO$_2$ fluxes from the treatment and control plots between January 2018 to January 2019 (see Table A1). Chamber measurements were carried out using permanently installed PVC collars with 0.4 m diameter and a height of 0.2 m, which were installed 0.15 m into the peat below

the *Astelia pumila* lawns in both treatments in October 2014. We identified and estimated plant species coverage within each collar (Table B2). We used a cylindrical, transparent chamber with a basal area of 0.12 m$^2$ and a height of 0.4 m to conduct soil–atmosphere flux measurements of CO$_2$ net ecosystem exchange (NEE). The chamber was equipped with a fan to ensure mixing of the headspace air, inlet and outlet ports to and from the infrared gas analyzer (IRGA; LI-840, Licor Inc., USA), a sensor (HOBO S-LIA-M003, Onset, USA) to measure photosynthetically active radiation ($PAR$), and a temperature sensor

(WatchDog 425, Spectrum Technologies, USA). CO$_2$ and water vapor concentration data were logged at three and $PAR$ at six second intervals on a data logger (CR-216 Campbell Scientific, Inc, USA). The chamber was connected by polyethylene tubing (2 mm inner diameter) to the IRGA. The gas analyzer was equipped with a pump transporting air from and to the chamber in a loop through the analyzer measurement cell. We avoided inducing a pressure pulse during chamber placement by equipping the chamber with a vent in its top cover. This opening was plugged immediately after the chamber was gently placed on a

collar for at least 3 minutes to conduct measurements. Collars were equipped with a water-filled rim to ensure a gas-tight seal between chamber and collar during measurements. Between measurements, the chamber was ventilated with ambient air until atmospheric background concentrations were measured inside the purged chamber. All measurements were performed under





a broad range of irradiance, starting early in the morning and continuing until late afternoon to encompass diurnal variations in CO$_2$ fluxes. At each collar, two or three consecutive measurements were performed with the transparent chamber, followed

by dark measurements, which were achieved by completely covering the chamber with a 5 mm thick reflective polyester fabric with aluminium coating on the outside. During dark measurements, GPP was assumed to be zero, and the measured CO$_2$ fluxes are therefore representative for ecosystem respiration, a combination of autotrophic and heterotrophic respiration.

## 2.5  CO$_2$ flux calculation

We calculated CO$_2$ fluxes using a routine (Eckhardt and Kutzbach, 2016) in MATLAB R2019a (The MathWorks Inc., USA)

that applies different regression models to describe the change in the chamber headspace CO$_2$ concentration over time and conducts statistical analysis. This application also includes a graphical interface and allows for the calculation of a flux with respect to a manually selected period of a time during chamber closure. We visually inspected all measurements with this tool and removed periods with perturbed gas concentrations which often occurred shortly after chamber placement. To avoid a large impact of the greenhouse effect during closure time, we furthermore selected relatively short periods after the initial

perturbations for flux calculation. As water vapor concentrations were measured simultaneously, we additionally selected periods during which water vapor concentrations did not go into saturation, potentially impacting plant activity and light transmittance of the chamber walls in case of condensation. The time periods used for flux calculation were therefore generally much shorter than the chamber placement time. The median flux calculation period length was 48 seconds for the 81 treatment fluxes as well as for the 153 control fluxes (see Figure A1 and Table A1). The comparably short time periods lead to close-

to-linear concentration increases over time. We therefore used linear regression in all cases to calculate a gas flux from the estimated slope parameter ($\frac{dc_{CO_2}}{dt}$) of the fitted line with Eqn. 1.

$$F_{CO_2} = \frac{1}{R} \frac{V}{A} \frac{p_{air}}{T_{air}} \frac{dc_{CO_2}}{dt} \tag{1}$$

where $F_{CO_2}$ is the molar flux of carbon dioxide (µmol m$^{-2}$ s$^{-1}$), $R$ is the ideal gas constant (8.3145 kg m$^2$ s$^{-2}$ mol$^{-1}$ K$^{-1}$), $V$ is the chamber volume (m$^3$), $A$ the collar area (m$^2$), $p_{air}$ is the air pressure (Pa) and $T_{air}$ is air temperature (K). Air

pressure (measured at nearby EC station) and temperature (measured inside chamber) as well as chamber volume and collar area are assumed to be constant during a flux calculation period. Prior to function-fitting, the measured CO$_2$ concentrations were referenced to the initial water vapour concentration and recalculated to $c_{CO_2,corr}$ by applying Eqn. 2.

$$c_{CO_2,corr} = \frac{c_{CO_2}}{1 - c_{H_2O}} (1 - c_{H_2O,init}) \tag{2}$$

where $c_{CO_2}$ and $c_{H_2O}$ are the concentrations of CO$_2$ and water vapor as measured with the IRGA and $c_{H_2O,init}$ is the initial

water vapor concentration at the beginning of the flux calculation period. We propagated the standard error of the slope parameter and uncertainty estimates for the constants (air pressure, collar area, headspace volume) and variables (water vapor concentration and air temperature) in the flux calculation equation through Eq. 1 to calculate the absolute standard error of the flux. The median relative flux errors of about 15 % were similar for treatment and control fluxes (see Figure A1).





## 2.6 CO₂ flux modeling

We modeled the measured CO$_2$ net ecosystem exchange fluxes (NEE, µmol m$^{-2}$ s$^{-1}$) from the treatment and control plots separately using a combination of two deterministic functions in a single bulk model as in Runkle et al. (2013). This model consists of a term modeling total ecosystem respiration (TER, µmol m$^{-2}$ s$^{-1}$) as an exponential air temperature ($T$) relation and a light saturation function (rectangular hyperbola) of $PAR$ to estimate photosynthesis (gross primary production GPP, µmol m$^{-2}$ s$^{-1}$). As dark measurements (dark respiration R$_d$, µmol m$^{-2}$ s$^{-1}$), where photosynthesis can be assumed to be 165 inactive, were available in our data set (treatment, n = 18; control, n = 34), we first estimated the respiration parameters $Q_{10}$ (dimensionless) and $R_{\text{base}}$ (µmol m$^{-2}$ s$^{-1}$) in Eqn. 3 (with the constants reference temperature $T_{\text{ref}}$ = 15 °C and $\gamma$ = 10 °C) from the dark flux measurements. Subsequently, we set these respiration parameter estimates as constants in the bulk model (Eqn. 4) and optimized the GPP parameters maximum photosynthesis $P_{\text{max}}$ (µmol m$^{-2}$ s$^{-1}$) and initial quantum yield $\alpha$ (dimensionless) using all fluxes including the ones measured with transparent chambers (treatment, n = 81; control, 170 n = 135). All parameter estimates were optimized using the SciPy Python library (Virtanen et al., 2020) by minimizing the squared model-data residuals. As the derived parameters can be interpreted as plant-specific, physiological properties due to our mechanistic modeling approach, we used them to characterize the treatment effect on plant properties.

$$R_{\text{d}}(T; R_{\text{base}}, Q_{10}) = R_{\text{base}} \times Q_{10}^{\frac{T - T_{\text{ref}}}{\gamma}} \tag{3}$$

$$\text{NEE}(PAR, T; P_{\text{max}}, \alpha) = \text{TER}(T) - \text{GPP}(PAR; P_{\text{max}}, \alpha) = R_{\text{base}} \times Q_{10}^{\frac{T - T_{\text{ref}}}{\gamma}} - \frac{P_{\text{max}} \times \alpha \times PAR}{P_{\text{max}} + \alpha \times PAR} \tag{4}$$

Eventually, we drove the optimized NEE models with half-hourly $PAR$ and $T$ records from our meteorological station at the site to calculate net CO$_2$ uptake of the treatment and control plots over the three main growing seasons (15 November to 15 March) 2016/2017, 2017/2018 and 2018/2019. We calculated uncertainty estimates of each modeled GPP, TER and NEE flux as 95 % confidence intervals based on Gaussian error propagation. The partial derivatives of Eq. 4 with respect to the parameters which we used in the course of error propagation are described in Holl et al. (2019). We simplified error propagation 180 by ignoring the random errors of temperature and radiation measurements which we assume to be rather small compared to the uncertainty of the flux estimates. We calculated the uncertainty of the annual sums by taking the square root of the sum of squared uncertainties of the individual 30 minute NEE sums.

## 3 Results

### 3.1 Treatment effects on temperature

The open side chamber treatment resulted in higher near-surface (1 cm above canopy) air temperatures at the treatment plots compared to the control plots. At the three replicate experiments which were equipped with temperature sensors, the mean air temperature difference over the measurement period of about one year was between 0.4 °C and 0.7 °C (see Figure D5). These





average annual differences are consistent over hourly raw data as well as daily and weekly averaged temperature measurements (see Figures D1 to D4). Soil temperatures at 10 cm depth were elevated in the same range only at two of three replicate

experiments. Annual means of daily and weekly averaged soil temperature differences were between 0.2 °C and 0.3 °C and 0.0 °C at the third plot.

As another way to characterize the temperature differences caused by the treatment, we calculated average diurnal cycles for single months and seasons (see supplementary Figures S1 to S32). We applied Mann-Whitney U-tests to compare the distribution means at each hour of day. We found significant ($p < 0.1$) mean air temperature differences between control

and treatment mostly at midday, between 13:00 and 15:00, coinciding with the solar radiation maximum. Midday maximum differences ranged between 1.2 °C and 3.0 °C in spring and summer and between 1.0 °C and 2.0 °C in autumn and winter. At other times of day, significant air temperature differences were only found at one of three plots during autumn and winter. The timing of significant seasonally averaged soil temperature (10 cm below the surface) differences at single hours of the day is less consistent over the replicate plots (compared to the timing of maximum air temperature differences). In summer

and at two of three plots, we found significant differences between midnight and early morning. Maximum soil temperature summer differences occurred between 03:00 and 05:00 at all three replicates. In winter, the comparably smaller temperature differences had significantly different means at each hour of day, while the diurnal course and the timing (11:00, 18:00, 22:00) of the maximum difference appears to be inconsistent over the three replicate plots. In spring and summer, maximum soil temperature differences lay in between -0.2 °C and 0.4 °C, meaning that at one plot, the treatment apparently led to a

soil temperature decrease. The same was the case in autumn and winter when maximum soil temperature differences ranged between -0.2 °C and 0.6 °C.

Results of the randomization test (see Figure D5) clearly show that the mean hourly temperature differences we measured between treatment and control plots were not only caused by random noise. The randomly generated mean temperature differences are centered around zero, and the probability of the actual mean difference to come from this generated distribution

is very low (One sample t-test: $p < 0.001$). We therefore conclude that the OSC treatment did increase surface as well as soil temperatures in the treatment plots significantly. In case of the soil temperatures the increases were, however, smaller and not consistent over the three replicates.

### 3.2 Treatment effects on leaf properties

In 2016, the average treatment leaf was significantly (Mann-Whitney U-test, $p < 0.01$) shorter in midsummer (January, Febru-

ary) and at the end of the growing season (April) compared to the control leaves. Within each treatment, leaf lengths were significantly (Kruskal-Wallis test, $p < 0.01$) different from each other in January, February and April 2016, indicating leaf growth. In contrast, at the beginning of the growing season (September 2017) of 2017/2018, treatment leaves tended to be longer than control leaves on average (Mann-Whitney U-test, $p < 0.1$). In the late growing season (March 2018) of 2017/2018, we could not find a significant difference ($p = 0.3$) between control and treatment leaf length means (see Figure 3).

Average leaf growth (see Figure B1) was neither significantly different (Mann-Whitney U-test, $p > 0.2$) between treatment and control plants in 2016 (13 January to 29 April 2016: 1.34 mm ± 2.21 mm (control); 1.28 mm ± 1.76 mm (treatment)) nor





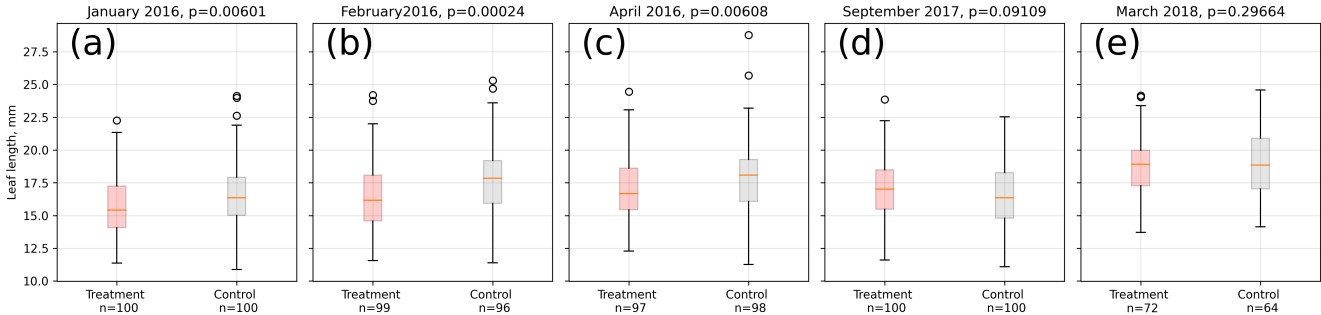

**Figure 3.** Comparison of *A. pumila* leaf lengths from treatment and control plots throughout the growing seasons 2015/2016 (panels a to c) and 2017/2018 (panels d and e). Mann-Whitney U-tests indicate highly significant ($p < 0.01$) differences between treatment and control leaf lengths in 2016. In September 2017, this difference is less significant ($p < 0.1$). In March 2018, leaf lengths did not differ between treatment and control plots.

**Table 1.** Estimated average growth rate of individual *Astelia pumila* leaves at the treatment and control plots in two seasons. Length measurements were taken on 13 January 2016, 29 April 2016, 25 September 2017 and 7 March 2018. Differences were divided by the number of days between observations to calculate growth rate.

| Season | Leaf growth per day, µm | | | | | |
| --- | --- | --- | --- | --- | --- | --- |
| | **Treatment** | | | **Control** | | |
| | n | Mean | SD | n | Mean | SD |
| **2015/2016** | 97 | 12 | 16 | 98 | 13 | 21 |
| **2017/2018** | 72 | 11 | 11 | 64 | 12 | 13 |

in 2017/2018 (25 September 2017 to 7 March 2018: 1.99 mm ± 2.20 mm (control); 1.77 mm ± 1.83 mm (treatment)). Again indicating plant development, one-sample t-tests and Wilcoxon signed rank tests show that the average growth is significantly different ($p < 0.0001$) from zero for both growing seasons and at the treatment and control plots. We calculated leaf growth
per day by normalizing the length increases with the observation period length for the respective season (see Table 1). This average leaf growth rate is with about 12 µm per day virtually constant for both treatments and in both seasons.

Leaf area differences were only significant ($p < 0.05$) in late summer indicating a larger mean leaf area at the control plots (see Figure 4). Leaf area means within the treatment group were not significantly different between midsummer and late summer. In contrast, the leaf area of the control plants was significantly ($p < 0.1$) larger in late summer compared to midsummer. It has
to be noted that the number of samples for these comparisons was limited; test results may therefore not be robust.

Results from our areal plant density estimation (see supplementary Table S1) suggest that areal plant density is significantly (Mann-Whitney U-test, $p < 0.05$) higher at the treatment plots compared to the control plots, whereas mean aboveground biomass per area is larger at the treatment plots but not significantly different ($p = 0.2$) from the control plots (see Figure 5). Simultaneously to the leaf length measurements in two growing seasons, we counted the total number of leaves per plant





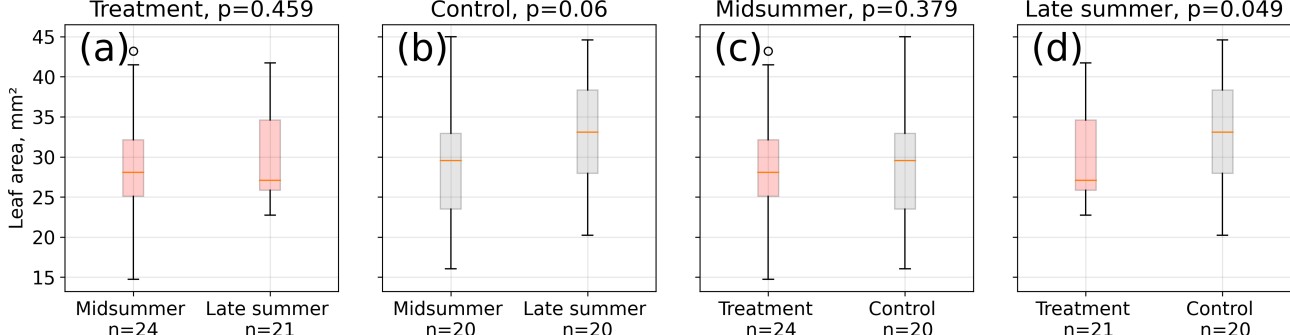

**Figure 4.** Leaf area measurements from 12 measurement days between 15 January 2016 and 04 March 2016. We divided the area estimates into two groups referring to midsummer and late summer (see supplementary Table S2) and compared the respective treatment and control means using a Mann-Whitney U-test. Average leaf area was not different between midsummer and late summer at the treatment plots (panel (a)), whereas control plot leaves (panel (b)) were on average longer in late summer than in midsummer, although at a low significance level (p < 0.1). Treatment and control plant average leaf area did differ significantly in late summer (panel (d), p < 0.05) but not in midsummer (panel (c), p = 0.4)

in January 2016 for 100 control and 100 treatment plants. The total number of leaves was 7 and equal for both groups (see supplementary Table S3). We did not detect differences in leaf water content, leaf dry mass, and specific leaf area between treatment and control plants (see supplementary Table S2).

We did not find significant differences in absolut leaf pigment contents between treatment and control plants at any of the measurement days (see supplementary Table S6). However, when considering the increase of pigments from the mid growing

season to its end, we found that on average control leaves increased chlorophyll *a* by a significantly (Mann-Whitney U-test, p < 0.05) larger amount between February and May 2016 than treatment leaves (see Figure 6).

Accordingly, treatment plants had the same number of leaves, which were, however, shorter (5 %) and with a smaller (10 %) surface area at the end of the season than control plants, whereas the number of plants per area is increased (10 %) at the treatment plots so that leaf biomass per area and specific leaf area is the same in both groups.

### 3.3 Treatment effects on CO$_2$ fluxes

We estimated bulk model parameters seperately for control and treatment fluxes as described in the methods section. With high coefficients of determination and with respect to the range of measured fluxes relatively low root mean square errors (RMSEs), model quality appears to be sufficient to yield meaningful parameters in all four cases (see Figures 7 and 8). Distinctions between treatment and control are most pronounced with respect to the parameter expressing the temperature sensitivity of

respiration (Q$_{10}$) and the parameter denoting the theoretical maximum photosynthesis at infinite photon supply (P$_{max}$). From the dark measurements (Figure 7), we estimated a nearly 80 % larger Q$_{10}$ at the treatment plots (control: 2.8; treatment: 5.0). At the same time, P$_{max}$, as estimated from the bulk model fit to chamber measurements (Figure 8), was more than 40 % lower




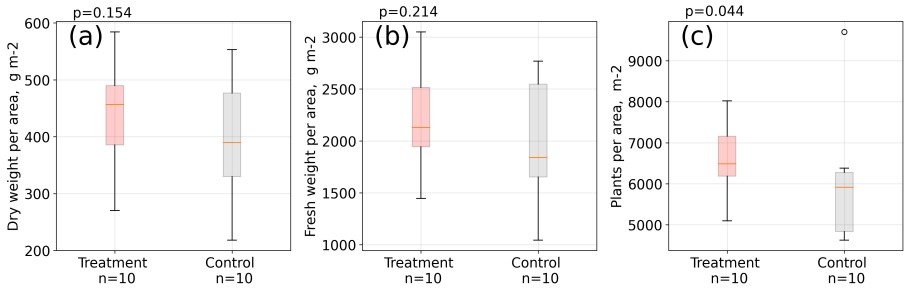

**Figure 5.** Aboveground *A. pumila* biomass. Dry weight (panel a) and fresh weight (panel b) per area and individual plants per area (panel c). Measurements were taken in April 2016 in about 10 cm by 10 cm large sampling rectangles within the control and treatment plots (see supplementary Table S1). About one and a half years after the installation of the open side chambers in October 2014, a significantly (Mann-Whitney U-test, p < 0.05) denser plant cover developed at the treatment plots compared to the control plots (panel c).

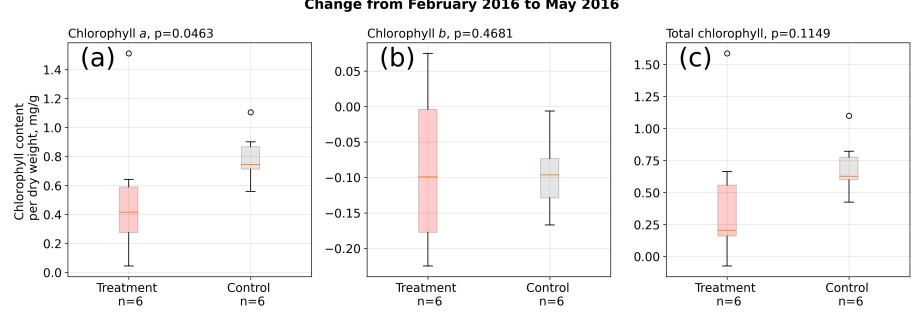

**Figure 6.** Difference of chlorophyll contents between February and May 2016 in *A. pumila* leaves from treatment and control plots. During this growing season, control plants appear to have increased their chlorophyll *a* content (panel (a)) to a significantly (p < 0.05) greater extent than treatment leaves. Sample size is, however, not large enough to firmly make the latter assertion.

at the treatment plots (control: 15.9 μmol m$^{-2}$ s$^{-1}$; treatment: 9.2 μmol m$^{-2}$ s$^{-1}$). We found less pronounced differences for base respiration $R_{\mathrm{base}}$ at 15 °C, which was elevated by 16 % at the treatment plots (control: 4.21 μmol m$^{-2}$ s$^{-1}$; treatment: 4.89 μmol m$^{-2}$ s$^{-1}$), and for the initial quantum yield $\alpha$, which was 23 % larger at the treatment plots (control: 0.031; treatment: 0.038). A comparison (see Figure C1) with bulk model parameter time series, which Holl et al. (2019) estimated from eddy covariance data at the same site for two years, reveals that our chamber-derived parameters are representative for the main growing season (15 November to 15 March). During this period, the control plot parameters from this study are in line with eddy covariance ecosystem scale estimates. Chamber flux measurements were biased towards the mid growing season (see Table A1 and Figure C1, panel (b)) where chamber-derived parameter estimates also overlap with eddy covariance parameter time series.

The fact that, at the treatment plots, the bulk NEE partitioning model is able to explain the comparably small CO$_2$ uptake in the high radiation range ($PAR > 1200$ μmol m$^{-2}$ s$^{-1}$, see Fig 8, panel B) further increases our confidence in the stepwise





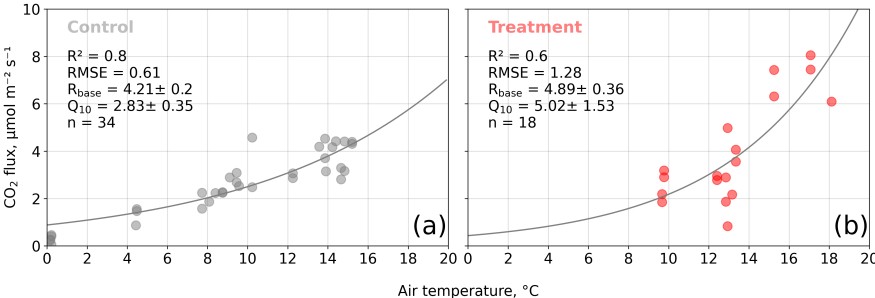

**Figure 7.** Dark respiration measurements acquired with opaque chambers at the control (panel (a)) and treatment (panel (b)) plots versus air temperature. We fitted an exponential function of air temperature (see Eqn. 7) to the measured carbon dioxide ($CO_2$) fluxes to estimate the ecosystem respiration parameters base respiration $R_{base}$ ($\mu$mol m$^{-2}$ s$^{-1}$) and temperature sensitivity $Q_{10}$ (dimensionless). Coefficients of determination ($R^2$) and root mean square errors (RMSEs) are given.

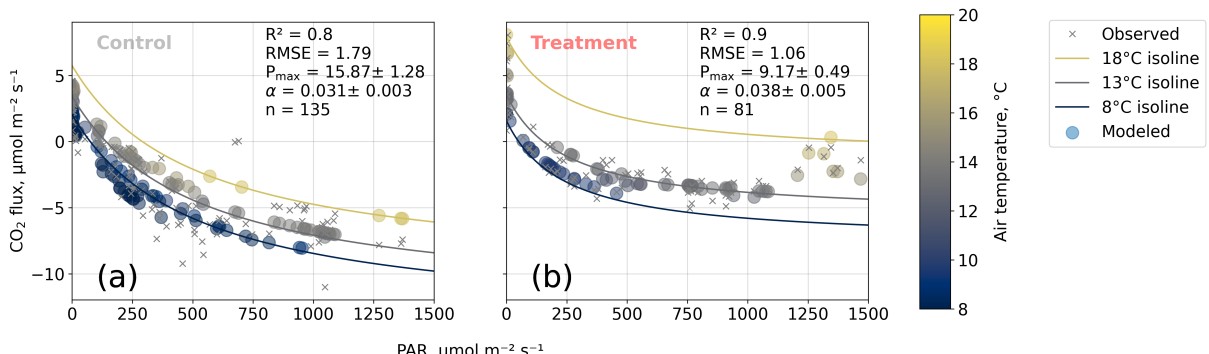

**Figure 8.** Net carbon dioxide ($CO_2$) flux versus photosynthetically active radiation PAR ($\mu$mol m$^{-2}$ s$^{-1}$). We modeled observed net ($CO_2$) fluxes from chamber measurements as a function of $PAR$ and air temperature with our bulk model (see Eqn. 4). Two of the four bulk model parameters ($Q_{10}$ and $R_{base}$) were determined based on dark respiration measurements (see Figure 7) and set as constants so that the bulk model was used to optimize the parameters maximum photosynthesis $P_{max}$ ($\mu$mol m$^{-2}$ s$^{-1}$) and initial quantum yield $\alpha$ (dimensionless). Coefficients of determination ($R^2$) and root mean square errors (RMSEs, $\mu$mol m$^{-2}$ s$^{-1}$) are given.

bulk model approach. Moreover, this behaviour highlights distinctive treatment effects on plant and soil traits: Photosynthesis
at the treated plots could not profit as much from higher radiation input as at the control plots while ecosystem respiration was promoted more by simultaneously increased temperatures so that net $CO_2$ uptake in high radiation and high temperature conditions is severely attenuated at the treatment plots. This temperature-dependent respiration increase can outweigh photosynthetic $CO_2$ uptake at high temperatures (see 18 °C isoline in Fig 8, panel (b)) across the whole radiation range.

     To investigate the treatment effect on growing season NEE sums, we drove the bulk models with half-hourly temperature and
radiation data measured at our site. We used data from the three main growing seasons 2016/2017, 2017/2018 and 2018/2019 between 15 November and 15 March. Due to a gap in meteorological data in 2019, we report only cumulative fluxes from





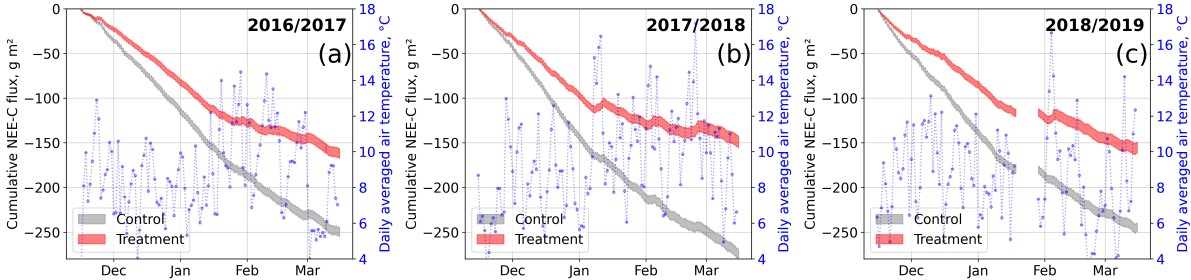

**Figure 9.** Modeled cumulative net ecosystem exchange (NEE) fluxes of $CO_2$ at the control and treatment sites, expressed as carbon (C) fluxes for three seasons. The cumulation period represents the main growing season from 15 November to 15 March. We used the previously determined bulk model parameters (see Figures 7 and 4) and observations of air temperature and photosynthetically active radiation to drive the models. Gaps in these time series in 2019 led to a gap in the cumulative curve in panel c. Areas represent the upper and lower bound of the model uncertainty.

the first two seasons here, data for 2018/2019 are plotted in Figure 9. The results document drastic differences of cumulative net $CO_2$ uptake between the control and treatment plots which are consistent over both considered main growing seasons. Model results suggest that the control plots sequestered between 55 % and 85 % more atmospheric $CO_2$-C (2016/2017: -250 ± 5 g m$^{-2}$; 2017/2018: -275 ± 5 g m$^{-2}$) than the treatment plots (2016/2017: -162 ± 5 g m$^{-2}$; 2017/2018: -149 ± 6 g m$^{-2}$). According to the data-based models, during three weeks in 2018 (07 January to 12 January, 01 February to 07 February and 20 February to 26 February), the treatment plots even turned from sinks into sources of atmospheric $CO_2$ as indicated by the upwards sloping cumulative curve during these especially warm periods.

## 4  Discussion

Our results show that the OSC treatment significantly increased air temperatures at the respective plots in the cushion bog of this study. As this method is inexpensive, causes limited soil disturbance, requires little maintenance and no power supply, it is especially suitable for remote sites and thus has commonly been applied in ecosystem warming experiments (e.g. Aronson and McNulty, 2009), also at sites with low vegetation like peatlands (Malhotra et al., 2020; Munir et al., 2015) or steppes (Liancourt et al., 2012; Sharkhuu et al., 2013). The temperature increases achieved by our OSC treatment are consistent with other passive warming experiments at high latitudes (Rustad et al., 2001; Zaller et al., 2009; Bokhorst et al., 2011). Compared to these studies, the increment of near-surface (1 cm above canopy) air temperature in our study (0.4 °C to 0.7 °C) was in the same range (Bokhorst et al., 2007; Prather et al., 2019), or smaller with respect to the results of Day et al. (2008) from Antarctica (1.3 °C to 2.3 °C) or the warming of 1.0 °C to 3.6 °C achieved in a tundra experiment (Biasi et al., 2008; Walker et al., 2006). Our OSC treatment led to significantly elevated air temperatures mainly during daytime as also observed in previous studies (Marion et al., 1997; Bokhorst et al., 2007) and to increased air temperatures during all four seasons. As opposed to air temperatures, our OSC treatment did not affect soil temperatures consistently over the replicate plots. The increases in soil





temperatures were generally lower compared to the air temperature increments. On the other hand, as shown by the results of the randomization test (see Fig. D5), soil temperature differences between treatment and control plots are unlikely to be caused by random processes only, indicating a, however inconsistent, treatment effect.

Apart from the desired temperature increase by reduced radiative energy loss, an OSC method can have secondary effects (Aronson and McNulty, 2009; Marion et al., 1997) like shading, altered micro-local wind patterns and attenuated turbulence or reduced radiation input. We are confident that in our experimental study, we were able to minimize these secondary effects by installing the semicircular plastic wall with its open half facing northwards. Due to the position of our site on the southern hemisphere, incoming radiation was only attenuated early in the morning and late in the afternoon, while the treatment plots

received direct radiation throughout most of the day. Moreover, we determined the light transmissivity of the wall material and found it to be between 80 % and 85 %. As winds most frequently came from north-northwestern directions, wind shelter effects caused by the OSCs were limited. At the fewer times when winds came from southern directions, sheltering by the OSC walls was most intense. However, the thereby attenuated turbulent energy transport did not consistently result in intensified warming accross the replicate plots as shown in supplementary Figure S33. At Plot 24, warming was slightly (about 0.1

°C) more intense during west-southwestern winds. At Plot 22, where warming was least effective overall, attenuated turbulent energy transport could explain the observed intensified warming during winds from the sheltered south-southwestern directions. Additionally, phases of north-northwestern wind directions at the same time were phases of relatively warm air temperatures (at 2 m, measured at EC station, see supplementary Figure S34), so that at Plot 25, warming was most intense during north-northwestern wind directions, when sheltering effects were most limited. A secondary effect we could not counteract lies in

the fact that the detected temperature differences were not consistent over the course of a day. Therefore, a small temperature increment at certain times of day and a larger difference at other times of day exposes the treatment plots not only to larger average temperatures but also to a larger temperature range.

    Between one and a half and three and a half years after the OSC treatment had been installed, we observed distinctions between the morphological features of the *A. pumila* plants in the control and treatment plots. Howevre, leaf growth (i.e.

length increase) did not differ significantly between treatment and control plants. In general, the growth rates we found are low but similar to values of other herbaceous vascular plants at comparable latitudes reported in the literature (Day et al., 2001; Rousseaux et al., 2001; Searles et al., 2002; Robson et al., 2003).

    Concurrently to the morphological alterations, we found that in treated plants certain physiological properties were modified. In particular, we observed a tendency for the treated plants to increase their chlorophyll *a* content less throughout the growing

season (see Figure 6) than control plants. At the treatment plots, a degradation in the efficiency of GPP, especially at high light levels, is documented by our bulk model parameter estimates showing reduced maximum photosynthesis and elevated initial quantum yield. Conversely, ecosystem respiration was enhanced considerably at the treatment plots. We could validate the bulk model parameter estimates we derived from plot-scale chamber measurements at the control plots with ecosystem-scale eddy covariance measurements from the same site. Additionally, the eddy covariance parameter estimates are given as time series

and thus enable an assessment of a period the chamber-derived parameters are representative for. We found that the stage of





plant development during the main growing season (15 November to 15 March) is best represented by the parameters reported in this study.

A large discrepancy exists between the temperature sensitivity parameters $Q_{10}$ at plot and ecosystem scale. The $Q_{10}$ estimate from eddy covariance is in line with values from most global ecosystems ($1.4 \pm 0.1$; Mahecha et al., 2010; Zhang et al., 2017)

but, in contrast to plot-scale measurements, this signal is representative for a mixture of surface types typical for cushion bogs including pools or moss-dominated patches. Especially due to the exclusion of pools, an elevated temperature sensitivity can therefore be expected on plot scale. However, a methodological bias could exist due to transient increases in leaf respiration when previously light-exposed plants are suddenly darkened as reviewed by Heskel et al. (2013). If such a bias exists, it would affect our stepwise bulk model approach (as our assumption that $R_d$ is equal to TER would not hold), and therefore could

alternatively explain the $Q_{10}$ discrepancy because in its first step, dark measurements are used to estimate the respiration parameters $Q_{10}$ and $R_{\mathrm{base}}$ (see Eqn. 3). On the other hand, we might have avoided disturbances caused by transient plant responses by excluding initially perturbed gas concentrations prior to flux calculation.

We tested if with a full four-parameter bulk model, which should largely remove the above mentioned bias as it does not rely on dark measurements, respiration parameter estimates differ from those derived with the stepwise approach (see Table C1).

Full bulk model $Q_{10}$ estimates of the treatment plots are still much higher ($3.65 \pm 0.64$) than the eddy covariance estimate but lower than the result when only using dark chamber measurements ($5.02 \pm 1.53$) to determine $Q_{10}$ and $R_{\mathrm{base}}$. However, the impact of choosing different sets of respiration parameters on the cumulative growing season TER sums for the three considered growing seasons is somewhat counterintuitive. Table C1 shows the respiration parameters we derived from the alternative modeling approaches and the relative differences in the cumulative growing season TER sums calculated with these

alternative models. In all six cases (three seasons, two treatments), the full four parameter bulk model TER estimate is between 10 % and 20 % larger than the stepwise estimate despite the larger $Q_{10}$ estimates from the stepwise approach. The reason for the larger TER sum from the model in which both parameters are smaller is that air temperature during the summation period often was below the reference temperature of 15 °C (see Eqn. 3). We therefore assume that when optimizing many model parameters at the same time, equifinality issues might be more important sources of uncertainty than transient respiration

increases of suddenly darkened plants. Moreover, the stepwise bulk model gives the more conservative TER estimate, making our conclusion of an increased respiration in the treatment plots less likely to be the result of an overestimation due to a methodological bias.

As discussed above, we are confident that an increased respiration at the treatment plots can be asserted. Due to the limited impact of the warming treatment on soil temperatures, we speculate that this increased total respiration could largely be at-

tributed to enhanced autotrophic respiration. Commonly (e.g. Chapin et al., 2011), upward bending $PAR$-NEE curves at high light levels (Fig. 8, panel (b)) can be explained by photooxidation, i. e. the (adverse) physical effect of high energy photons on plant tissue. In a warming experiment in a sub-Arctic (68 °N) heath community, Bokhorst et al. (2010) found that warming caused stress and enhanced lipid peroxidation resulting in cell and tissue damage in treatment plants. In our study, the *A. pumila* plants under warming treatment could have been more prone to photooxidation due to warming-induced stress.





A possible explanation for the simultaneous increase in respiration along with a diminished efficiency of photosynthesis has been outlined by Brooks and Farquhar (1985) and by Dusenge et al. (2019). The authors found that photorespiration can be enhanced at elevated temperatures due to the decreasing ability of the enzyme RuBisCO to distinguish between $CO_2$ and molecular oxygen ($O_2$) with increasing temperatures. The — for the purpose of photosynthetic efficiency preferable — carboxylation reaction ($CO_2$ fixation) can thereby be hampered while the oxygenation reaction ($CO_2$ release) is intensified.

The result of such a mechanism would match our observations of the elevated temperature sensitivity of respiration and the diminished GPP at the treatment plots. On the other hand, the temperature difference archived in our treatment experiment might have been too small to cause the observed differences by the above-described dependency of RuBisCO characteristics on temperature. The GPP differences between treatment and control plots in this study are additionally connected to the observed changes in leaf area and leaf chlorophyll *a* content in the course of the growing season which were both larger at the

control plots. The differences in the latter plant traits might have driven GPP variability primarily or in conjunction with the temperature dependence of RuBisCO properties. An alternative or additional explanation for the increased respiration at the treatment plots might be connected to *A. pumila* root dynamics which we did not observe in this study. Cushion plants develop a high belowground-to-aboveground biomass ratio (Fritz et al., 2011) and a relatively large, dense root and rhizome system (Kleinebecker et al., 2008). Root growth might have been intensified by the warming treatment, as observed in the study of

Malhotra et al. (2020), and could have led to enhanced respiration.

In our study, warming was paralleled by alterations of morphological and physiological cushion plant properties. Treated *A. pumila* plants formed denser cushions consisting of smaller individual plants; their photosynthetic capacity was hampered while respiration intensified leading to a drastically diminished growing season $CO_2$-C net sink at warmed *A. pumila* cushions. Other warming studies found similar as well as contrary effects of experimental warming on plant properties. In general,

vegetation responses to warming are highly species-specific as noted by Prather et al. (2019). Studying the impact of warming on maritime antarctic plant communities, Bokhorst et al. (2007) found that open plant communities (grasses and lichens) were more negatively affected by warming than densely growing plants like mosses and dwarf shrubs. Similarly, Walker et al. (2006) report that woody plants benefited from warming at 11 sites across the tundra biome. Contrary to Bokhorst et al. (2007), Walker et al. (2006) show that moss and lichen cover decreased as a result of warming. As also indicated at our site by a longer average

treatment leaf length at the onset of the growing season (see Fig. 3, panel (d)), Livensperger et al. (2016) report accelerated plant development at passively warmed plots in the initial growing season from a *Eriophorum*-dominated tussock tundra site on the North Slope of Alaska. While the tussock community development was accelerated in spring, overall growing season plant growth was not increased, matching our observations at the cushion bog. Furthermore, Bokhorst et al. 2010 identified spring-like plant development in winter in warmed plots at a sub-Arctic (68°N) heathland as the major driver of warming-induced

tissue damage which manifests as lipid peroxidation and impacts metabolic efficiency. In general, cold and wet ecosystems appear to be more sensitive to warming than drier ecosystems (Peñuelas et al., 2004). At microform level, a similar observation was made by Munir et al. (2015) with respect to growing season $CO_2$-C net uptake in a *Sphagnum*-dominated bog in Canada (55°N). The authors found that at drier hummocks net $CO_2$ uptake was elevated by warming whereas at wetter hollow sites,





more $CO_2$ was emitted. Likewise, Hopple et al. (2020) report an increase of dark respiration with temperature based on a

warming experiment in a *Sphagnum*-dominated, treed, low shrub bog in Minnesota (47°N).

## 5 Conclusions

We conducted a warming experiment in a southern hemisphere cushion bog to investigate the morphological and physiological responses of the cushion-forming plant *Astelia pumila* to elevated temperatures as they are projected to occur on the southern hemisphere in a future climate. We found morphological distinctions, with *A. pumila* plants growing in denser cushions and

having shorter leaves at the treatment plots. We detected no difference in specific leaf area, but control leaves had on average a larger end-of-season leaf area. Furthermore, plant physiology was apparently affected by the warming treatment as well. To derive plant physiological parameters, we conducted a total of 234 chamber $CO_2$ flux measurements and fitted mechanistic models for photosynthesis and ecosystem respiration driven by radiation and air temperature to the flux observations from the treatment and control plots. We found that at elevated temperatures, *A. pumila* photosynthesis was less efficient whereas

ecosystem respiration was enhanced. Due to the limited effect of our experiment on soil temperatures, we attribute the total respiration increase largely to an increase in autotrophic respiration. We propose an increase in photorespiration as a response to warming as one likely underlying mechanism since it could explain the diminished gross primary production and enhanced respiration simultaneously. Apart from alterations of the photosynthetic apparatus, differences in leaf morphology and chlorophyll contents between treatment and control plants most likely additionally, or even decisively, contributed to the observed

GPP variability. Respiration variability could additionally have been impacted by changes in root respiration and stress-induced enhanced photooxidation.

Driving our NEE model with meteorological data from two growing seasons revealed that the *A. pumila* cushions where temperature conditions were manipulated cumulatively took up 55 % and 85 % less $CO_2$-C than the cushions of unaltered control plots over the main growing seasons with respect to two exemplary years. At warm days with high irradiance, the net

$CO_2$ sink character of the treated cushions was lost. The characteristics of *A. pumila* plants at the treatment plots were altered so that at very warm temperatures (> 18 °C) photosynthetic $CO_2$ uptake does not compensate for respiratory $CO_2$ loss across the entire radiation range.

*Author contributions.* VP and LK conceptualized and administered the planning of the research activity and acquired the funds for it. VP, MFC and JE conducted the field work. VP and DH conducted literature research. DH and VP analyzed the data, created visualizations and

wrote the original draft. VP, DH and LK reviewed and edited the original draft.

*Competing interests.* No competing interests are present.



*Acknowledgements.* For facilitating and greatly supporting our field work we want to cordially thank Wiebke Münchberger, Till Kleinebecker, Christian Blodau, Mauro Britos Navarro and Norman Rüggen. We thank Prefectura Naval Argentina for granting us permission to work on their property and for kindly inviting us to benefit from their facilities at the field site.

This work was supported by the Cluster of Excellence CliSAP (EXC177), Universität Hamburg, funded by the German Research Foundation (DFG), by the CONICET-MINCyT-DFG international cooperation grant 3829/15 by grant PIDUNTDF 15/16 of Universidad Nacional de Tierra del Fuego, Antártida e Islas del Atlántico Sur (UNTDF) and by the DFG project KU 1418/6-1.





## Appendix A: Flux measurement statistics

**Table A1.** Number of chamber $CO_2$ flux measurements we conducted between January 2018 and January 2019 at the treatment and control plots of our site.

| Date | Number of flux measurments | | |
|------|-------|-----------|---------|
| | Total | Treatment | Control |
| 23 January 2018 | 11 | 0 | 11 |
| 01 February 2018 | 24 | 0 | 24 |
| 06 March 2018 | 23 | 23 | 0 |
| 07 March 2018 | 60 | 36 | 24 |
| 14 March 2018 | 26 | 0 | 26 |
| 15 May 2018 | 11 | 0 | 11 |
| 16 July 2018 | 16 | 0 | 16 |
| 28 November 2018 | 12 | 12 | 0 |
| 29 November 2018 | 15 | 0 | 15 |
| 17 January 2019 | 18 | 10 | 8 |
| 18 January 2019 | 18 | 0 | 18 |
| Sum | 234 | 81 | 153 |

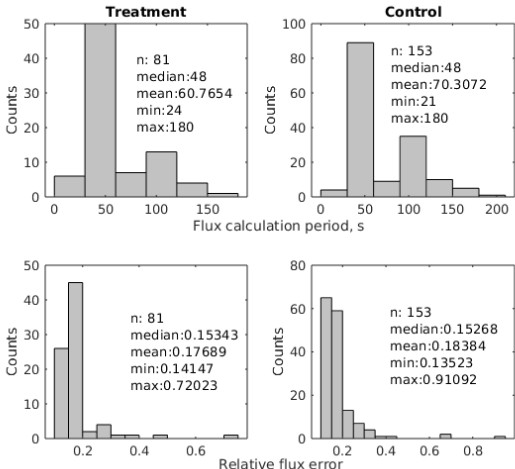

**Figure A1.** Flux calculation statistics. Of the three minutes chamber closure time, a median period of 48 seconds was selected for flux calculation. Flux uncertainties mostly amounted to between 10 % and 20 % of the respective flux.





## Appendix B: Site conditions and vegetation properties

**Table B1.** Comparison of the meteorological conditions at our study site during three consecutive seasons. Growing degree days (GDD) are defined as the sum of all positive differences between daily average temperatures and reference temperatures of 5 °C ($GDD_5$) and 10 °C ($GDD_{10}$). 2016/2017 was the warmest year on average and warm days (largest $GDD_5$) occured most consistently. Additionally, this year's summer was very rainy. 2017/2018 had by far the most very warm days (largest $GDD_{10}$), especially in summer, and was also the driest year.

| Season | Daterange | Meanair temperature,°C | Precipitation sum,mm | $GDD_5$, °C | $GDD_{10}$, °C |
|---|---|---|---|---|---|
| 2016/2017 | WholeYear (Septemberthrough August) | 6.6 | 793.2 | 822 | 77 |
| 2017/2018 | | 6.3 | 532.8 | 787 | 122 |
| 2018/2019 | | 5.9 | 670.4 | 643 | 62 |
| 2016/2017 | Spring (Septemberthrough November) | 7.2 | 129.6 | 217 | 9 |
| 2017/2018 | | 6.1 | 134.8 | 149 | 14 |
| 2018/2019 | | 5.6 | 133.4 | 140 | 6 |
| 2016/2017 | Summer (Decemberthrough February) | 9.1 | 303.4 | 370 | 54 |
| 2017/2018 | | 9.9 | 188.0 | 442 | 93 |
| 2018/2019 | | 8.8 | 201.8 | 305 | 42 |
| 2016/2017 | Autumn (Marchthrough May) | 7.0 | 195.8 | 211 | 14 |
| 2017/2018 | | 6.3 | 146.2 | 173 | 14 |
| 2018/2019 | | 6.4 | 233.2 | 177 | 14 |
| 2016/2017 | Winter (Junethrough August) | 3.2 | 164.4 | 24 | 0 |
| 2017/2018 | | 2.9 | 63.8 | 23 | 0 |
| 2018/2019 | | 2.6 | 102.0 | 21 | 0 |
| 2016/2017 | Maingrowingseason (15November to15March) | 8.7 | 380.2 | 456 | 60 |
| 2017/2018 | | 9.5 | 263.4 | 540 | 104 |
| 2018/2019 | | 8.6 | 239.2 | 409 | 54 |





**Table B2.** Estimated plant coverage at three treatment plots and three control plots where gas flux measurements were performed.

| | Estimated coverage, % | | | | | | |
|---|---|---|---|---|---|---|---|
| | **Treatment** | | | **Control** | | | |
| **Plant species** | Plot 1 | Plot 2 | Plot 3 | Plot 4 | Plot 5 | Plot 6 | Plot 7 |
| *Astelia pumila* | 82 | 85 | 89 | 57 | 57 | 88 | 89 |
| *Caltha dioneifolia* | 8 | 5 | 2 | 6 | 2 | 5 | 3 |
| *Gaultheria pumila* | 7 | 6 | 2 | 24 | 18 | 0 | 2 |
| *Myrteola nummularia* | 0 | 4 | 6 | 8 | 0 | 6 | 1 |
| *Drosera uniflora* | 2 | 0 | 0 | 0 | 0 | 0 | 0 |
| *Donatia fascicularis* | 1 | 0 | 0 | 4 | 18 | 2 | 0 |
| *Sphagnum magellanicum* | 0 | 0 | 1 | 0 | 0 | 0 | 0 |
| *Empetrum rubrum* | 0 | 0 | 0 | 2 | 0 | 0 | 0 |
| *Tetroncium magellanicum* | 0 | 0 | 0 | 0 | 5 | 0 | 1 |
| *Nanodea muscosa* | 0 | 0 | 0 | 0 | 0 | 0 | 1 |
| *Nothofagus betuloides* | 0 | 0 | 0 | 0 | 0 | 0 | 3 |

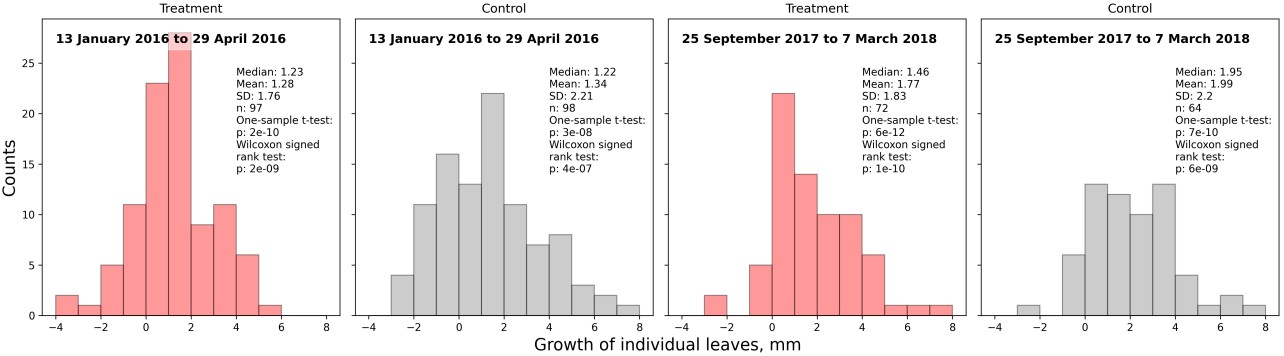

**Figure B1.** Distribution of leaf growth estimates for *Astelia pumila* at the control and treatment plots. Growth estimates are calculated as differences of length measurements of leaves which we individually tracked throughout the season. We attribute the occurrence of negative growth values to random measurement uncertainty of the length measurements with a digital caliper. One-sample t-tests and Wilcoxon signed rank tests from the SciPy Python library were used to check if the average growth values were significantly different from zero, i. e. growth took place.





**Appendix C: Comparison of NEE model parameter estimates obtained with different models and with different measurement techniques**

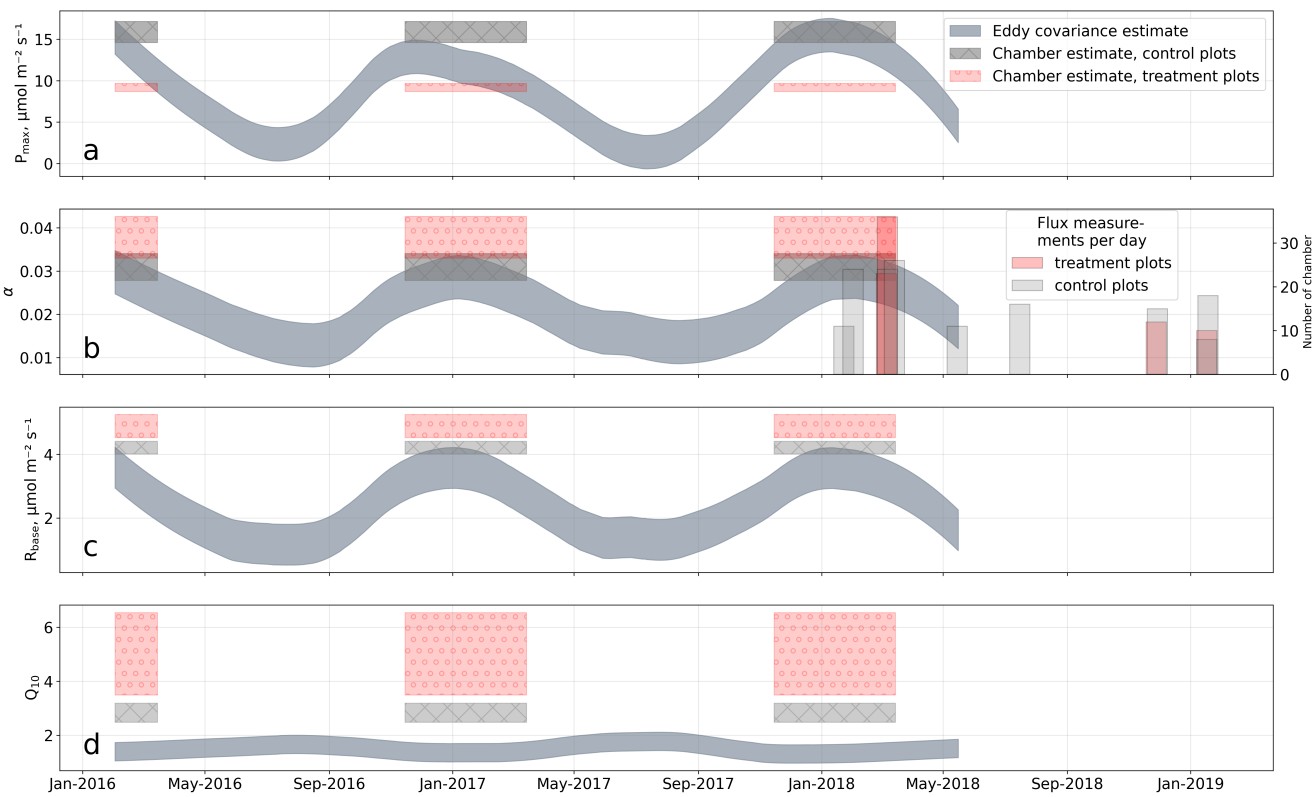

**Figure C1.** Comparison of bulk model (see equation 4) parameters from this study with parameter time series derived from eddy covariance measurements by Holl et al. (2019) for the same site. Panel (a) shows photosynthetically active radiation $P_{max}$, panel (b) initial quantum yield $\alpha$ (dimensionless), panel (c) base respiration $R_{base}$ and panel (d) temperature sensitivity of respiratoin $Q_{10}$. The comparison allows for an estimation of a period within the seasonal course the chamber-derived parameters are representative for. We found that the parameter estimates from this study best represent conditions during the main growing season to which our chamber measurements are biased towards (see secondary y-axis in panel (b))





**Table C1.** Respiration parameters base respiration $R_{base}$ (µmol m$^{-2}$ s$^{-1}$) and temperature sensitivity $Q_{10}$ derived from optimizing the full (four parameter) bulk model (Eqn. 4) compared with the estimation from dark measurements (Eqn. 3) within our stepwise bulk modeling approach. Additionally, the impact of choosing differently derived parameter sets on the cumulative total ecosystem respiration sums $\sum TER_{full}$ and $\sum TER_{step}$ over the main growing seasons (15 November to 15 March) of three examplary years for which air temperature records exist for our site are shown. Full bulk model TER sums for the treatment and control plots are between 10 % and 20 % larger than stepwise bulk model TER sums which we used to calculate carbon dioxide net ecosystem exchange sums (see Figure 9) in this study.

|  |  | Full bulk model | Stepwise bulk model | $\frac{\sum TER_{full}}{\sum TER_{step}}$ |
|---|---|---|---|---|
| Treatment | $R_{base}$ | $4.92 \pm 0.30$ | $4.89 \pm 0.36$ | 1.17 (2016/2017) |
|  | $Q_{10}$ | $3.65 \pm 0.64$ | $5.02 \pm 1.53$ | 1.12 (2017/2018) |
|  |  |  |  | 1.16 (2018/2019) |
| Control | $R_{base}$ | $2.96 \pm 0.32$ | $4.21 \pm 0.20$ | 1.19 (2016/2017) |
|  | $Q_{10}$ | $1.12 \pm 0.13$ | $2.83 \pm 0.35$ | 1.09 (2017/2018) |
|  |  |  |  | 1.19 (2018/2019) |



## Appendix D:  Time series of air and soil temperatures inside and outside the warming treatment

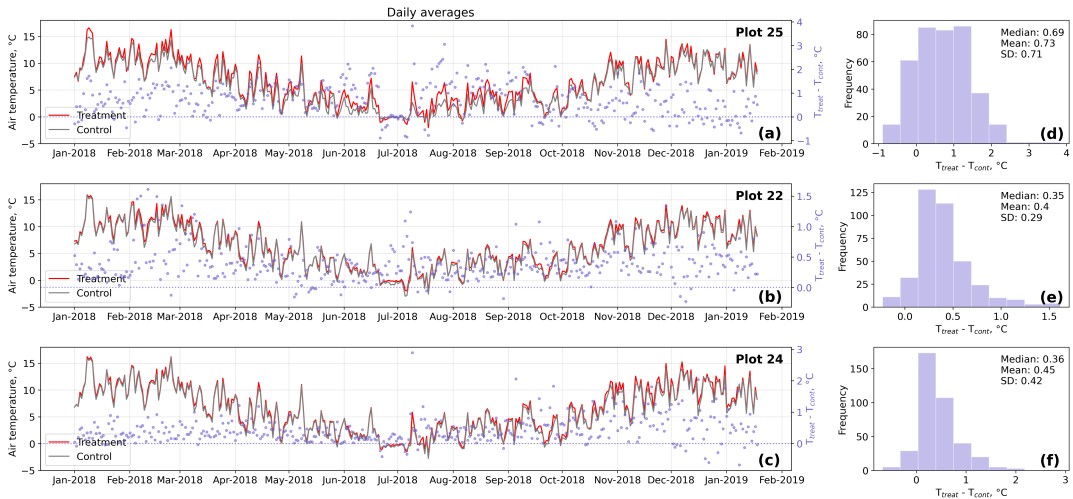

**Figure D1.** Time series (panels (a) to (c)) of daily averaged hourly air temperature measurements inside ($T_{\mathrm{treat}}$) and outside ($T_{\mathrm{cont}}$) warming treatment plots. The difference between $T_{\mathrm{treat}}$ and $T_{\mathrm{cont}}$ are mostly positive. The distributions of the temperature differences including mean, median and standard deviation (SD) are shown in panles (d) to (f).

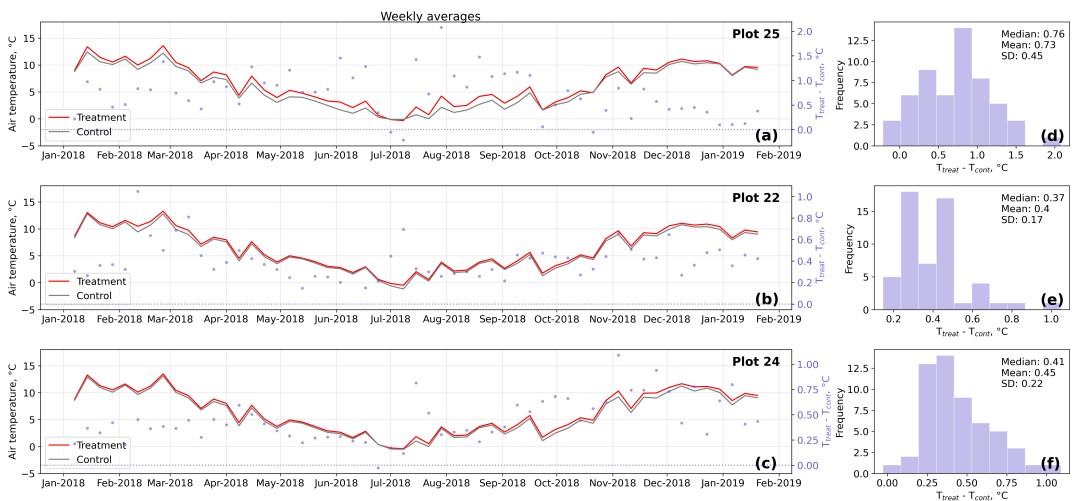

**Figure D2.** Time series (panels (a) to (c)) of weekly averaged hourly air temperature measurements inside ($T_{\mathrm{treat}}$) and outside ($T_{\mathrm{cont}}$) warming treatment plots. The difference between $T_{\mathrm{treat}}$ and $T_{\mathrm{cont}}$ are mostly positive. The distributions of the temperature differences including mean, median and standard deviation (SD) are shown in panles (d) to (f).



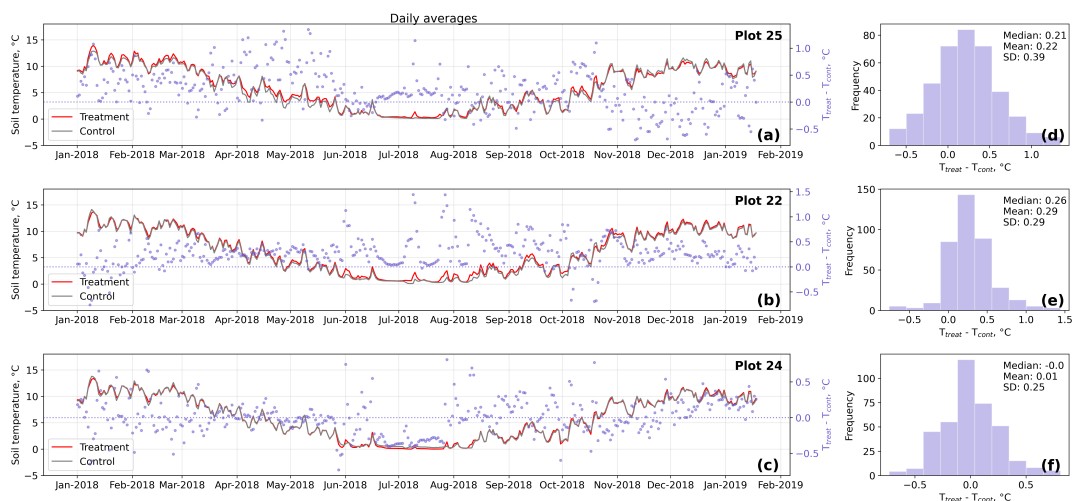

**Figure D3.** Time series (panels (a) to (c)) of daily averaged hourly soil temperature measurements inside ($T_{\text{treat}}$) and outside ($T_{\text{cont}}$) warming treatment plots. The difference between $T_{\text{treat}}$ and $T_{\text{cont}}$ are mostly positive. The distributions of the temperature differences including mean, median and standard deviation (SD) are shown in panles (d) to (f).

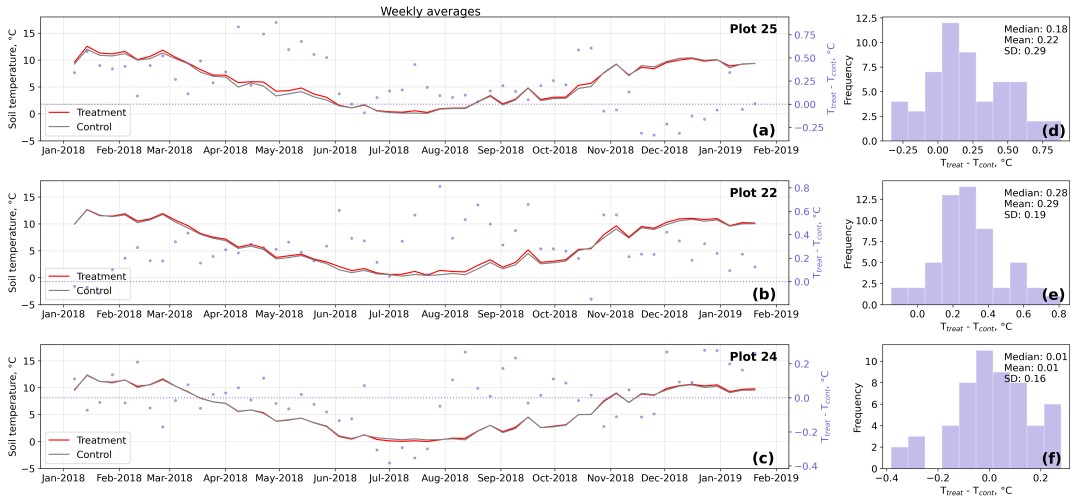

**Figure D4.** Time series (panels (a) to (c)) of weekly averaged hourly soil temperature measurements inside ($T_{\text{treat}}$) and outside ($T_{\text{cont}}$) warming treatment plots. The difference between $T_{\text{treat}}$ and $T_{\text{cont}}$ are mostly positive. The distributions of the temperature differences including mean, median and standard deviation (SD) are shown in panles (d) to (f).





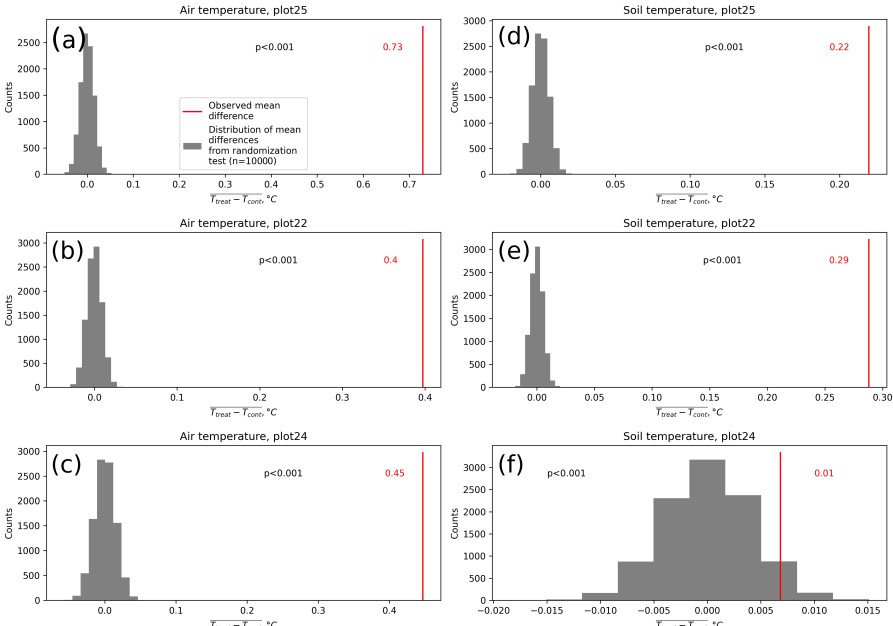

**Figure D5.** Results of the randomization test we performed to check if the average differences between temperatures at the treatment and control plots are likely to be caused by random noise rather than a systematic distinction between the conditions inside and outside the treatment plots.



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
