# Peer review of "Cushion bog plant community responses to passive warming in southern Patagonia"

_Biogeosciences, 2020_

## Short Comment (SC1) · 4 Feb 2021

Comment on the under-review manuscript "Cushion bog plant community responses to passive warming in Southern Patagonia" by: Erika Hodgson*, Naomi Gunasekara*, Jonathan Garrido-Mirapeix Munn*, Ryan Newman*, Kai Westwell*, Georgios Kazanidis** *Undergraduate student in the course "Critical Thinking in Ecological and Environmental Sciences" at the University of Edinburgh **Tutor in in the course "Critical Thinking in Ecological and Environmental Sciences" at the University of Edinburgh

Dear authors, as part of the undergraduate course "Critical Thinking in Ecological and Environmental Sciences" at the University of Edinburgh we have read carefully the mentioned above manuscript and we would like to express here our thoughts. We

have found this piece of work is, in overall, a timely and interesting manuscript and we hope that our thoughts will help the authors to improve the status of their under-review paper.

Abstract We feel that that the abstract is well written with a use of clear language. We think, however, that the authors should highlight/elaborate further on some of their interesting findings. E.g., we think that they should have highlighted that while some aspects of the plant's biology were affected by the rising temperatures (e.g., biometric features, photosynthesis, respiration) others (e.g., pigments) were not; suggestions explaining this "divergence" would be welcome. Also, we feel that the authors should have clarified which IPCC climate scenarios they have incorporated in their work.

Introduction The introduction has a good structure in overall. We think, however, that the authors could have addressed the particular research gap more thoroughly. For example, the authors should have explained why they have chosen to work with the species Astelia pumila e.g., is this a cosmopolitan species, does it have a key role in ecosystem structure and functioning? Clarification on these aspects could increase the overall impact of the manuscript and its findings and make it more accessible. We feel also that the "Introduction" would have benefited if the authors had made some null hypothesis about the impacts of the changing abiotic parameters on the species biology. We feel that the lasts parts of the Introduction should mention to the readers which are the main aims and objectives of the work and how the findings will fit into the larger picture. Currently the last parts of the Introduction (e.g., lines 41-45) should be removed to the "Materials and Methods" section of the paper. Materials and Methods Lines 89-90: Please mention the measurements units used for measuring the size of the plant's leaves. Line 96: Can you please explain how the number "86" has been reached/calculated? The current number of replicates (n=3) for the semicircular plastic walls is acceptable; however higher numbers could provide higher statistical robustness. We acknowledge that logistical constraints may have prevented the use of these plastic walls. There was also no mention of the number of individual organisms present

per plot. Line 113: The sampling dates were mentioned, but it was not said how many replicate measurements were taken for the the $CO_2$ flux measurements on the treatment and control plots during this time. We do acknowledge that this information was in appendix A1.

Results Lines 185-210 ("Treatments effects on temperature"). We think that these lines would fit better in the "Materials and Methods" section. Lines 215-218 suggest that the growing season ranges from September to April. Please clearly define the range of the growing season in the "Materials and Methods" section. Figure 3, panel d (September 2017). Based on the p-values that are provided the differences are not statistically significant ($p > 0.05$). The authors state in the caption that "...this difference is less significant ($p < 0.1$)" which sounds a bit odd. Table 1. Please mention in the caption of the Table the measurement units for the growth rate. Figure 4. Some of the information provided in the caption is rather redundant (e.g. We divided the area estimates into two groups referring to midsummer and late summer (see supplementary Table S2) and compared the respective treatment and control means using a Mann-Whitney U-test). We feel that this level of details is not necessary in here and would be adequate if it is only shown in the "Materials and Methods", section.

Discussion We feel that it is not the best way to start a Discussion by highlighting technical aspects; instead the authors could have given a succinct overview of the major/most interesting findings based on which they will build their Discussion.

Studying the impacts of rising temperature on plant biology is a key feature; however, it is common knowledge that it is not the parameters that exerts stress on organisms; based on that it would be welcome if authors mention that a multiple-stressor experiment would have provided a better insight about the effects of climate change. Also, elaboration (even a succinct one) on the findings of other relevant studies about the impacts of multiple stressors on plant performance and implications about ecosystem structure and functioning (e.g., elemental cycling) would be great.

Conclusions The last part of the conclusions should have highlighted how the key findings of the present work fit into the bigger picture e.g. in the functioning and resilience of ecosystems where these plants are abundant. It may be beneficial to conclude by highlighting why your findings are relevant and potentially suggest management strategies to reduce the impact on Astelia pumila. It would be welcome also the authors to highlight some research gaps that would simulate future research works.

---

## Referee Comment (RC1) · Tariq Munir (Referee) · 21 Feb 2021

General comments:

Climate warming has been frequently reported to occur under future climate change scenarios; therefore, it is very important and interesting to evaluate the impacts of it on various ecosystems – specifically those with significant accumulations of peat following a long-term carbon sink functioning since Holocene. One of the most important carbon sink ecosystems with peat accumulation is peatland – could be various kinds. Thus, this study of investigating carbon flux responses to expected warming in a cushion bog is important and interesting and may help advise ecosystem management and related policy development. This manuscript merits consideration for publication once

the reviewer's comments are appropriately addressed to improve the quality of this rigorous research.

Specific comments: The abstract section is quite comprehensive and written with great clarity and brevity. The reader can feel the weakness of the study at line 6 where passive warming is shown to be only 0.4-0.7 with a very small n=3. No SD or SE is provided. For example, the uncertainty may be larger than the warming itself, and then what would be the strength of this study – not sure. But let me see ahead in the results. Additionally, it is amazing to find that small temperature increases of 0.4-0.7 switched the system to a large increase in respiration and/or 55-85% decreases in CO2-C sequestration (sink to source).

Line 43-45: It is noticeable that authors referred to old experimental warming research works conducted in 1997 and 2009 with chambers, and more recent works are ignored here (One of the references is a review (Aronson and McNulty, 2009), not an experiment; however, it works well here). For example, Lyons et al. 2020 (Journal of Vegetation Science), Strack et al. 2019 (Ecohydrology), Yang et al. 2017 (Atmospheric Environment), Munir et al. 2017 (Ecohydrology). Also, the last paragraph of this manuscript describes how the chamber used in this study works, while this should be a separate comprehensive paragraph with more and recent reviews in the methods section.

Line 113: Again, more recent references could be added here for readers to gain knowledge breadth.

Line 114: I am worried about how authors adjusted the volume of the chamber when the chamber was placed on the collar with protruding vascular vegetation which certainly occupied some volume of the chamber headspace and should be adjusted while applying PV=nRT in the respective excel column/empirical modelling – not just the coverage. Did authors adjust the transmittance in their chamber flux calculations? Additionally, this experiment used open side chambers for passive warming – what

happened to rainfall or precipitation inputs compared to the control plots is not even briefly mentioned. Mention clearly if the chamber was open on top as well.

Line 122: describe IRGA (make, model, location).

Line 143-146: I appreciate that the authors used linear regression; an increase in $CO_2$ concentration in the chamber headspace should be linear if the chamber volume is large and is corrected for inside temperature and pressure following ideal gas law. I see the measurement chamber size is very small (0.12 m2 basal area x 0.4 m height) compared to the contemporary research works that used passive warming with open top or side chamber – I know measurement time is somewhat smaller in this research. Also, covering the in-situ soil and vegetation has been frequently reported to manipulate the spontaneous $CO_2$ fluxes across the soil-vegetation-air continuum (Hanson et al., 1993; Davidson et al., 2002; Denmead and Reicosky, 2003; Kutzbach et al., 2007) – did authors made any adjustments to mitigate these effects or interferences?

I think the statistical analysis section is missing – I need to know what design of experiment authors used, did they applied repeated measures with a mixed model or what statistical analysis techniques were used.

Line 187: SD?

Line 194: what is $p<0.1$?

Fig. 3: axes labels are very small fonts.

Fig. 7. Why can't the y-axis have the label "ecosystem respiration" instead to let the reader know what flux is this?

Fig. 9. It is unusual to see different flux measurement units used for $CO_2$ measurement in a single experiment.

Discussion: Line 295 onwards: how did researchers exclude the effect of rainfall or precipitation – since rainfall is from the top, not sides. How did rainfall enter the experimental chamber area? If the top was also open, then why it was not described here. I feel the authors need to describe the warming chamber in more detail in the setup of the experiment section. The authors have included a full paragraph on the wind direction effect. Please add a wind direction diagram, for example, the wind rose in the methods section to substantiate your arguments in the discussion section. Adding this figure may substantially strengthen the authors' arguments. I know wind may not be in a single direction like northerly but could take other directions as well with varying speeds during a year.

Line 355: space.

I notice authors keep switching between CO2 and CO2-C, please be consistent.

Line 398: please provide the temperature range here in brackets.

Line 404: replace treatment plots with warmed plots.

Line 412-413: replace "the A. pumila cushions where temperature conditions were manipulated" with "warmed A. pumila cushions".

Line 416: please do not mix manipulated warming and weather temperatures – be clear in conclusions. As authors used here >18°C, they can use the passive warming values as well to help the reader a take-home message with exact passive warming and their calculated or modelled effects on GPP and Respiration.

Authors may want to provide passive warming average temperatures during study years/seasons at various plots in a separate table.

---

## Referee Comment (RC2) · Gerald Jurasinski (Referee) · 21 Mar 2021

Dear Authors,

thank you very much for this well structured and well written manuscript. Please apologise my delayed review. I should have checked in early than I would have known, that this is an easy one. Along most of the manuscript it is a very interesting read that is strongly rooted in references and addresses an interesting and timely issue (change in C turnover in a warming world) with a comparably easy to implemented approach (open side chambers). In some parts of the methods and the results section it gets a bit lengthy but most of the provided information is interesting anyway (but sometimes maybe not necessary). The striving for providing the completest possible picture is also

reflected in a huge number of supplemental materials and without suggesting explicitly to skip this or that I would suggest the authors discuss among themselves which of the materials are really worth reporting here.

I made comments and suggestions on a pdf that you find attached. Apart from minor comments and/or suggestion regarding phrasing and wording or very few typos or grammar issues I suggest mainly slight edits or request some more information in the method section regarding the open side chamber approach and ask you to provide a little bit more information on the timing issue (e.g., not all measurements were carried out at the same time, why some measurements were spread across three months in the starting year but later not anymore, how do you think may the measuring on control and treatment plots in differently timed campaigns (at least partially) have influenced your findings). An illustration that explains the sampling design in a graphical manner, especially telling what was measured when would be great.

Further, I am missing a subsection of the discussion section on "limitations" (although there are some aspects n this regard intermingled in the discussion) in which you could for instance discuss the last point mentioned in the parentheses above. Finally, I would like to see a paragraph in the conclusion that discusses what your results mean for these peatlands, the world, future studies. Instead the conclusion is bloated with summary stuff that doesn't need to be there since its a conclusion not a summary.

I do not address all single issues here since you can find them in the commented pdf.

After all, in my opinion, this is a very interesting and well developed contribution that only needs some editing before it can be finally published in Biogeosciences.

Please also note the supplement to this comment:
https://bg.copernicus.org/preprints/bg-2020-440/bg-2020-440-RC2-supplement.pdf

**Supplement:**

[revised manuscript text omitted]

---

## Author Comment (AC1) · 10 May 2021

**Author reply to Referee comments from **Referee # 1 Tariq Munir** from 21 February 2021 ([https://doi.org/10.5194/bg-2020-440-RC1](https://doi.org/10.5194/bg-2020-440-RC1)) on:**

**Cushion bog plant community responses to passive warming in southern Patagonia**

Verónica Pancotto et al.

Reviewer comments (RC)
Author comments (AC)
Mentioned line numbers refer to the originally submitted manuscript
Manuscript changes (MC)

General comments: Climate warming has been frequently reported to occur under future climate change scenarios; therefore, it is very important and interesting to evaluate the impacts of it on various ecosystems – specifically those with significant accumulations of peat following a long-term carbon sink functioning since Holocene. One of the most important carbon sink ecosystems with peat accumulation is peatland – could be various kinds. Thus, this study of investigating carbon flux responses to expected warming in a cushion bog is important and interesting and may help advise ecosystem management and related policy development. This manuscript merits consideration for publication once the reviewer's comments are appropriately addressed to improve the quality of this rigorous research.

Specific comments: The abstract section is quite comprehensive and written with great clarity and brevity. The reader can feel the weakness of the study at line 6 where passive warming is shown to be only 0.4-0.7 with a very small n=3. No SD or SE is provided. For example, the uncertainty may be larger than the warming itself, and then what would be the strength of this study – not sure. But let me see ahead in the results. Additionally, it is amazing to find that small temperature increases of 0.4- 0.7 switched the system to a large increase in respiration and/or 55-85% decreases in CO2-C sequestration (sink to source).

We realize that in all its brevity the abstract can also be misunderstood. The reported range refers to the average difference between warmed plot and control plot at three replicates. We decided to report the range because we do not think it is very informative to calculate the standard deviation/error of three values. We dedicate sections in the results and discussion parts to characterize the temperature time series in more detail and apply statistical methods to investigate if the observed temperature differences at the three replicate plots could also be the result of random noise (which they are not at high significance levels).

We replaced "(n=3)" in the abstract with:

(At the three of ten treatment plots which were equipped with temperature sensors)

Line 43-45: It is noticeable that authors referred to old experimental warming research works conducted in 1997 and 2009 with chambers, and more recent works are ignored

here (One of the references is a review (Aronson and McNulty, 2009), not an experiment; however, it works well here). For example, Lyons et al. 2020 (Journal of Vegetation Science), Strack et al. 2019 (Ecohydrology), Yang et al. 2017 (Atmospheric Environment), Munir et al. 2017 (Ecohydrology). Also, the last paragraph of this manuscript describes how the chamber used in this study works, while this should be a separate comprehensive paragraph with more and recent reviews in the methods section.

At this stage of the manuscript (the end of the introduction), we only mention the method while we indeed address the experimental setup in more detail in subsection 2.2 "Setup of warming experiment" in the methods section.

We incorporated the newly suggested references and changed the paragraph from line 38 to 45 to:

To partly simulate future conditions, warming studies have commonly been conducted. Passive methods to manipulate soil and air temperatures have been chosen in studies focusing on high latitude peatlands (Laine et al., 2019; Lyons et al., 2020; Mäkiranta et al., 2017; Munir et al., 2017; Strack et al., 2019; Zaller et al., 2009) as these methods are cost-effective and appropriate for remote sites with limited power supply. Passive warming devices like open top chambers (OTCs) act as "solar energy traps" (Marion et al., 1997) primarily by reducing radiative heat loss (Aronson and McNulty, 2009). We conducted a field experiment to determine how cushion- forming plants respond to moderate experimental warming. We manipulated the temperature conditions passively with open side chambers (OSCs) similar to the ITEX Corners presented by Marion et al. (1997).

Line 113: Again, more recent references could be added here for readers to gain knowledge breadth.

More references added, see above.

Line 114: I am worried about how authors adjusted the volume of the chamber when the chamber was placed on the collar with protruding vascular vegetation which certainly occupied some volume of the chamber headspace and should be adjusted while applying PV=nRT in the respective excel column/empirical modelling – not just the coverage. Did authors adjust the transmittance in their chamber flux calculations? Additionally, this experiment used open side chambers for passive warming – what happened to rainfall or precipitation inputs compared to the control plots is not even briefly mentioned. Mention clearly if the chamber was open on top as well.

The vegetation establishes a dense cover, 2 to 3 cm above the soil surface, as shown in Figure 2. We estimated the chamber headspace individually for each plot a by taking into account surface irregularities as well as vegetation and collar height. PAR during measurements was measured inside the chamber. As clarified before, the device had an open-top design and its opening was facing north. We did not measure precipitation inside the plots, however, the water table (see Tables S3 to S5 in supplementary material) was near the surface throughout the vegetation periods guaranteeing plant water supply. We added in line 55:

Cushion-forming plants establish a very dense and low (about 2 to 3 cm above the soil surface) vegetation canopy (Figure 2).

In line 115, we added:

We estimated the head space, considering the distance from the vegetation to the border of the collar plus the height of the chamber above the collar.

Also, we added "internally" in line 117:

The chamber was internally equipped with...

Line 122: describe IRGA (make, model, location).

Which IRGA was used is specified in line 118.

Line 143-146: I appreciate that the authors used linear regression; an increase in $CO_2$ concentration in the chamber headspace should be linear if the chamber volume is large and is corrected for inside temperature and pressure following ideal gas law. I see the measurement chamber size is very small (0.12 m2 basal area x 0.4 m height) compared to the contemporary research works that used passive warming with open top or side chamber – I know measurement time is somewhat smaller in this research. Also, covering the in-situ soil and vegetation has been frequently reported to manipulate the spontaneous $CO_2$ fluxes across the soil-vegetation-air continuum (Hanson et al., 1993; Davidson et al., 2002; Denmead and Reicosky, 2003; Kutzbach et al., 2007) – did authors made any adjustments to mitigate these effects or interferences?

Actually the gas flux chamber we used was rather large. It is for example more than ten times larger in volume compared to commercially available systems offered by Licor Biosciences (https://www.licor.com/env/products/soil_flux/specs-chambers.html). The combination of a large volume and a short closure time (less than 50 seconds on average, see Figure A1) resulted in near linear concentration increases. Adjustments made to counteract disturbances and interferences of the gas flux caused by measuring it include: 1) usage of a large chamber, 2) a short chamber closure time, 3) removal of initial pertubations prior to flux calculation.

I think the statistical analysis section is missing – I need to know what design of experiment authors used, did they applied repeated measures with a mixed model or what statistical analysis techniques were used.

Since $CO_2$ fluxes are highly dynamic due to their radiation dependence, we decided to compare average data-derived response functions (Equation 4) and not control and treatment fluxes (or their means) directly. Our sampling concept was to try to measure fluxes at the treatment and control plots during a wide range of light and temperature conditions in order to be able to estimate the parameters of the response functions for the different plot types as accurately as possible. As the main (non-linear) drivers of net $CO_2$ flux are assumed to be known in this approach and are implemented in the response function, we did not apply any method to identify flux drivers like mixed linear models.

Line 187: SD?

See our reply to the comment to the abstract. In our opinion the range of the three values is more informative than reporting a standard deviation.

Line 194: what is $p < 0.1$?

P-value of the Mann-Whitney U test as described in preceding sentence.

Fig. 3: axes labels are very small fonts.

Font size increased

Fig. 7. Why can't the y-axis have the label "ecosystem respiration" instead to let the reader know what flux is this?

Ecosystem respiration is the process leading to CO2 flux; CO2 flux is the quantity, we think the label is correct. For clarity we changed it to: Respiration CO2 flux, µmol m-2 s-1

Fig. 9. It is unusual to see different flux measurement units used for CO2 measurement in a single experiment.

We chose this unit for easier comparability with literature values.

Discussion: Line 295 onwards: how did researchers exclude the effect of rainfall or precipitation – since rainfall is from the top, not sides. How did rainfall enter the experimental chamber area? If the top was also open, then why it was not described here. I feel the authors need to describe the warming chamber in more detail in the setup of the experiment section. The authors have included a full paragraph on the wind direction effect. Please add a wind direction diagram, for example, the wind rose in the methods section to substantiate your arguments in the discussion section. Adding this figure may substantially strengthen the authors' arguments. I know wind may not be in a single direction like northerly but could take other directions as well with varying speeds during a year. Line

The OSC has no cover, rainfall can enter, see Figure 2. Polar histograms describing the frequency of wind directions (Figures S33 and S34) are shown in the supplementary material and are referred to in the main text in this paragraph (lines 304 and 308).

355: space. I notice authors keep switching between CO2 and CO2-C, please be consistent.

It is useful to talk about CO2-carbon for comparison with literature values and with other components of the carbon cycle. Ecosystem carbon balances and long-term carbon accumulation rates of peatlands are commonly reported as carbon and not as CO2 balances (e.g. Yu et al., 2010; Loisel and Yu, 2013; Bunsen and Loisel, 2020).

Line 398: please provide the temperature range here in brackets.

The conclusions were completely restructured the temperature range is now also mentioned here. Also see our response to the comment of referee #2 to line 397.

Over the main growing season of two exemplary years, warmed A. pumila cushions cumulatively took up 55 % and 85 % less CO2-C than the cushions of unaltered control plots. This change in net C uptake is considerable, especially when comparing the amount of artificial warming achieved in our experiment (annual average between 0.4 °C and 0.7 °C at the three of ten replicates which were equipped with temperature sensors) with temperature projections for the region from the Coupled Model Intercomparison Project Phase 6 (CMIP6).

Line 404: replace treatment plots with warmed plots.

Done

Line 412-413: replace "the A. pumila cushions where temperature conditions were manipulated" with "warmed A. pumila cushions".

Done

Line 416: please do not mix manipulated warming and weather temperatures – be clear in conclusions. As authors used here >18C, they can use the passive warming values as well

to help the reader a take-home message with exact passive warming and their calculated or modelled effects on GPP and Respiration.

We suspect a misunderstanding here. What we want to say is: Warming alters plant properties and leads to a stronger temperature response of NEE from treated plants. As mentioned above, we rewrote the conclusions entirely.

Authors may want to provide passive warming average temperatures during study years/seasons at various plots in a separate table.

We do, see Appendix D.

**References**

Bunsen MS, Loisel J.: Carbon storage dynamics in peatlands: Comparing recent- and long-term accumulation histories in southern Patagonia. Glob Chang Biol. , 26(10), 2020

Loisel, J. and Yu, Z.: Holocene peatland carbon dynamics in Patagonia, Quaternary Sci. Rev., 69, 125–141, 2013

Yu, Z., Loisel, J., Brosseau, D. P., Beilman, D. W., and Hunt, S. J.: Global peatland dynamics since the Last Glacial Maximum, Geophys. Res. Lett., 37, L13402, 2010

Lyons, C. L., Branfireun, B. A., McLaughlin, J., & Lindo, Z. (2020). Simulated climate warming increases plant community heterogeneity in two types of boreal peatlands in north–central Canada. Journal of Vegetation Science, 31(5), 908-919.

Mäkiranta, P., Laiho, R., Mehtätalo, L., Straková, P., Sormunen, J., Minkkinen, K., ... & Tuittila, E. S. (2018). Responses of phenology and biomass production of boreal fens to climate warming under different water-table level regimes. Global Change Biology, 24(3), 944-956.

Munir, T. M., Khadka, B., Xu, B., & Strack, M. (2017). Mineral nitrogen and phosphorus pools affected by water table lowering and warming in a boreal forested peatland. Ecohydrology, 10(8), e1893.

Strack, M., Munir, T. M., & Khadka, B. (2019). Shrub abundance contributes to shifts in dissolved organic carbon concentration and chemistry in a continental bog exposed to drainage and warming. Ecohydrology, 12(5), e2100.

Yang, G., Wang, M., Chen, H., Liu, L., Wu, N., Zhu, D., ... & He, Y. (2017). Responses of CO2 emission and pore water DOC concentration to soil warming and water table drawdown in Zoige Peatlands. Atmospheric Environment, 152, 323-329.

Zaller, J. G., Caldwell, M. M., Flint, S. D., Ballaré, C. L., Scopel, A. L., & Sala, O. E. (2009). Solar UVB and warming affect decomposition and earthworms in a fen ecosystem in Tierra del Fuego, Argentina. Global Change Biology, 15(10), 2493-2502.

---

## Author Comment (AC2) · 10 May 2021

**Author reply to Referee comments from **Referee # 2 Gerald Jurasinski** from 21 March 2021 ([https://doi.org/10.5194/bg-2020-440-RC2](https://doi.org/10.5194/bg-2020-440-RC2)) on:**

**Cushion bog plant community responses to passive warming in southern Patagonia**

Verónica Pancotto et al.

Reviewer comments (RC)
Author comments (AC)
Mentioned line numbers refer to the originally submitted manuscript
Manuscript changes (MC)

Dear Authors,
thank you very much for this well structured and well written manuscript. Please apologise my delayed review. I should have checked in early than I would have known, that this is an easy one. Along most of the manuscript it is a very interesting read that is strongly rooted in references and addresses an interesting and timely issue (change in C turnover in a warming world) with a comparably easy to implemented approach (open side chambers). In some parts of the methods and the results section it gets a bit lengthy but most of the provided information is interesting anyway (but sometimes maybe not necessary). The striving for providing the completest possible picture is also reflected in a huge number of supplemental materials and without suggesting explicitly to skip this or that I would suggest the authors discuss among themselves which of the materials are really worth reporting here.

We moved plots S9 to S32 from the supplements to this manuscript to the supplements of our data submission at Pangaea.

I made comments and suggestions on a pdf that you find attached. Apart from minor comments and/or suggestion regarding phrasing and wording or very few typos or grammar issues I suggest mainly slight edits or request some more information in the method section regarding the open side chamber approach and ask you to provide a little bit more information on the timing issue (e.g., not all measurements were carried out at the same time, why some measurements were spread across three months in the starting year but later not anymore, how do you think may the measuring on control and treatment plots in differently timed campaigns (at least partially) have influenced your findings). An illustration that explains the sampling design in a graphical manner, especially telling what was measured when would be great.

We added a chart to the supplementary material further illustrating the timing of our different field campaigns, see below. Basically, the treatment was installed in 2014; 2 years later, leaf lengths measurements and samples for lab analyses were collected and another two years later, samples were again collected and chamber measurements were performed. What we did when mostly had logistical reasons, gas analyzers or lab time are not always available, remote roads are not safe to drive on in winter etc.. We do discuss the bias of our flux measurements towards the growing season (line 259 and caption of Appendix Figure C1). Also see responses to line comments below.

Further, I am missing a subsection of the discussion section on "limitations" (although there are some aspects n this regard intermingled in the discussion) in which you could for instance discuss the last point mentioned in the parentheses above. Finally, I would

like to see a paragraph in the conclusion that discusses what your results mean for these peatlands, the world, future studies. Instead the conclusion is bloated with summary stuff that doesn't need to be there since its a conclusion not a summary.

We completely rewrote the conclusions, see below. The section now includes a further statement about the main limitation of our study with our measurements being biased towards the growing season. As mentioned by the referee, the discussion includes various aspects which potentially further limit the explanatory power of our study.

I do not address all single issues here since you can find them in the commented pdf.

After all, in my opinion, this is a very interesting and well developed contribution that only needs some editing before it can be finally published in Biogeosciences.

**Line comments (extraced from pdf):**

Line 20: Move reference to the end of the sentence

Done

Line 20: increased. „Enhance" carries with it the notion of „making it better"

Agreed. We replaced "enhanced" with "intensified" to avoid repetition of "increased".

Line 22: Isn't that true also in other ecosystems?

Added "Similar to other ecosystems, " to beginning of sentence

Line 26: „Increased compared to the soil matrix"

Changed to: "...increased compared to the surrounding soil matrix"

Line 28: Provide examples of species names please

Examples added in parenthesis.

Line 28: in

Done

Line 36: This start would profit from a linking word like „Additionally" or something similar

Done

Line 39: on morphological and physiological traits of cushion plants

Done

Line 39: Because it is

Whole paragraph changed in response to comment from Referee #1

To partly simulate future conditions, warming studies have commonly been conducted. Passive methods to manipulate soil and air temperatures have been chosen in studies focusing on high latitude peatlands (Laine et al., 2019; Lyons et al., 2020; Mäkiranta et al., 2017; Munir et al., 2017; Strack et al., 2019; Zaller et al., 2009) as these methods are cost-effective and appropriate for remote sites with limited power supply. Passive warming devices like open top chambers (OTCs) act as "solar energy traps" (Marion et al., 1997) primarily by reducing radiative heat loss (Aronson and McNulty, 2009). We conducted a field experiment to determine how cushion- forming plants respond to moderate experimental warming. We manipulated the temperature conditions passively with open side chambers (OSCs) similar to the ITEX Corners presented by Marion et al. (1997).

Line 54: I know, it is probably correct but I have always a slight issue with below ground parts reaching „up to".. Shouldn't it be „down to"

"Up to" refers to the range of values here. 2 m is the largest value. We think this sentence is unambiguous, no change made.

Line 57: …low cover, in total not more than about 20% areal cover
Done

Line 74: But still the plants get 20 to 16% less radiation. Couldn't this influence your results?
Not that much due to the position of the chamber, see discussion, line 295.

Line 87: So the temperatures were measured later, right? After all, the temperature series and the leaf property measurements where not done at the same time. Would be good to discuss this directly in the methods and offer an explanation.
We added a chart to make the timing of the different measurements clearer. We refer to the figure after line 10:
See supplementary Figure S35 for an overview of the timing of our different sampling and measurment campaigns between 2014 and 2019.

Lenght measurements (LM)
Leaf area (LA)
Aboveground biomass (AB)
Photosynthetic pigments (PP)
Density (D)
chamber CO2 flux measurements (CO2)

Line 93: Why so many months for the start point?
We are not talking about start points here. In the season 2015/2016, we measured three times (beginning, middle and end of growing season). In the season 2017/2018, we measured two times (beginning and end of growing season).

Line 96: A figure, i.e. an illustration, how you did it, would be great
Gannt chart added to supplements, see above. Sampling described in more detail, see below.

Line 100: For this you have to take leafs home but I didn't see anything mentioning your field sampling for this. Please add
True, we made a mistake here. We changed the beginning of the paragraph starting in line 96 to:
All leaves we sampled for lab analysis were put in plastic ziplock bags, transported to the lab and stored in a refrigerator until the next day if they were not processed the same day. During 12 measurement days between 15 January 2016 and 04 March 2016 (see supplementary Table S2), we sampled in total 86 sun-exposed, fully expanded leaves. We took pictures of the leaves which included a ruler so we could estimate their area using the software ImageJ (Rueden et al., 2017).

Line 103: Maybe the same leaves were used for the above described analysis? Be it as it may, I would like to see some information on the sampling.

No, this was a different sampling campaign. Information on sampling added, see above.

Line 109: I have to do in keeping up with what was done when.. Here the above mentioned illustration to explain the sampling and analysis approach could help readers following

Gantt chart added, see above.

Line 112: So obviously together with the temperature measurements? This collars, don't they have a strong effect on the vegetation and C cycling when they are permanently installed?

Yes, temperature measurements were performed during the same timespan like the flux measurements.

We would argue the other way around: Because the collars were installed permanently four years (!) before measurements started the disturbance of soil and vegetation by collar installation was long enough ago to not affect carbon cycling decisively.

Line 124: Was it plugged immediately after placing or immediately after finishing the 3 minute measurement. That is not clear from the phrasing here.

Agreed, sentences were rearranged for clarification.

The openig was plugged immedtiately after the chamber was placed on the collar.

Line 138: I know that many people are doing this. But it introduces a subjective element in to this flux estimation...

We agree, it does add a subjective element. On the other hand, it is much harder to train an algorithm to identify pertubations than just to use human judgement/feature detection capabilities. From our point of view, the possibility of visual inspection is rather a feature than a shortcoming of this software/user interface. For us, this was also a way to use as many measurements as possible from this remote site from where it is not easy to get more data.

Line 143: Why so many more control than treatment fluxes?

We do not think that a large number of control fluxes affects data analysis negatively in any way. This imbalance in N between treatment and control fluxes is not the result of considerations with respect to experimental design. Rather, practical reasons are responsible, like the fact that not all flux measurements taken during this campaign are related to and part of this manuscript. Another practical reason could be that we tried to cover a large radiation range with our flux measurements. It possibly took us longer to get enough measurements for the control plots.

Line 189: Interesting. Do you have an explanation for this? Something to do with water in the ground?

Suspicion added:

We suspect that sensor placement might have been suboptimal, and the temperature probe was not installed as deep as at the other plots.

Line 191: Is this really enough to draw generalisable conclusions? I am not questioning this in general but would like some discussion on this in the methods section

No conclusion here, we merely report averages. Not entirely sure if we understand the comment and/or what it refers to.

Line 194: Clearly understandable because of the way you implemented the treatments
Yes, we expected higher differences at midday.

Line 205: Is this really necessary for the research question to discuss this in this much detail?
Yes, we think a thorough description of the effects of the warming treatment belongs to the clarification of the research question.

Line 207: Avoid such judgemental statements in general and especially in the results section.
Ok, "clearly" removed.

Line 231: Couldn't this also feed back on the leaf lengths (shorter lengths with denser growth)?
Exactly, that is what we are saying in the summary of this section, see line 242.

Line 244: was
Ok, both occurences of "is" changed to "was".

Line 246: Not necessary
Ok, removed.

Caption of Figure 6: Not entirely clear what is shown in the different panels. Further: Delete the title of the plot. You tell readers what they see in the figure caption
Ok, caption changed to:
Change of chlorophyll contents from February to May 2016 in A. pumila leaves from treatment and control plots. During this growing season, control plants appear to have increased their chlorophyll a content (panel (a)) to a significantly ($p < 0.05$) greater extent than treatment leaves. Differences in the contents of chlorophyll b (panel (b)) and total chlorophyll (panel (c)) did not change significantly. Sample size is, however, not large enough to firmly make these assertions.

Caption of Figure 8: I really like this figure although the 18°C isoline is really weakly determined in the treatment
Yes, we agree the 18 °C isolines are quite weakly determined at the treatment and control plots.

Line 274: If something suggests something it is a good indication that the statement should rather be place in the discussion section
This is clearly a result. We removed the beginnig of the sentence ("Model results suggest that").

Line 314: However
Done

Line 339: „would differ"?
Done

Line 379: Very likely this strongly depends on the morphologies of the plants under study
Exactly, as stated in following sentence.

Line 382: The question is whether over a bit longer periods this would just lead to changes in density and/or abundance??
Yes, we agree. We are, however, not able to answer this question. We are merely reporting results from other studies here.

Line 386: From an Eriophorum…
Done

Line 397: Try to shorten and put as start of next sentence, like „In our warming experiment…" Because it is not a summary, its a conclusion
We agree that there are too many summary elements in the conclusions in general. However, we think one introductory sentence summarizing the research question at the beginning of the conclusions is appropriate and serves as a reminder for the reader. We therefore left this sentence unchanged while we completely rewrote the rest of the conclusions.

We conducted a warming experiment in a southern hemisphere cushion bog to investigate responses of the cushion-forming plant Astelia pumila to elevated temperatures as they are projected to occur on the southern hemisphere in a future climate. At warmed plots, A. pumila grew in denser cushions and had shorter leaves leading to unchanged aboveground biomass per area. Furthermore, A. pumila physiology was altered so that at warmed plots, photosynthesis was less efficient while respiration was intensified. We propose an increase in photorespiration as a response to warming as one likely underlying mechanism since it could explain the diminished gross primary production and enhanced respiration simultaneously. Apart from alterations of the photosynthetic apparatus, differences in leaf morphology and chlorophyll contents between treatment and control plants most likely additionally, or even decisively, contributed to the observed GPP variability. Respiration variability could additionally have been impacted by changes in root respiration and stress-induced enhanced photooxidation.

Over the main growing season of two exemplary years, warmed A. pumila cushions cumulatively took up 55 % and 85 % less $CO_2$-C than the cushions of unaltered control plots. This change in net C uptake is considerable, especially when comparing the amount of artificial warming achieved in our experiment (annual average between 0.4 °C and 0.7 °C at the three of ten replicates which were equipped with temperature sensors) with temperature projections for the region from the Coupled Model Intercomparison Project Phase 6 (CMIP6). Estimates for contrasting Shared Socioeconomic Pathways (SSPs) show increases in mean annual 2 m air temperature of 1 °C (SSP1-2.6) and 2 °C (SSP5-8.5) from 2014 to 2100 (Wieners et al. 2019a, b). In conjunction with our findings, a considerable weakening of the long-term C sink strength of austral cushion bogs in a future climate seems likely. However, the temporal cover of flux measurements in our study was biased towards the growing season and more data from the shoulder seasons and winter, when temperatures are lower but photosynthesis of the evergreen A. pumila is ongoing, would be desirable and should be collected in future studies.

Line 401: Again summary elements. Keep statements to what you conclude from your findings.

Agreed, see comment above

Line 417: Add one paragraph on what this means for this kind of ecosystems under future expected warming

See restructured conclusions above

Caption of Table A1: Could this measuring differently in control and treatment plots have an influence on your findings regarding the differences in $CO_2$ fluxes? Please discuss in the discussion section.

We do not think that the timing of flux measurements or the greater number of control fluxes affected data quality negatively. In our sampling design and due to the high temporal variability of $CO_2$ fluxes, measuring fluxes over a wide range of light and temperature conditions was prioritized over measuring at control and treatment plots on the same day or equally often. Also see our response to the comment to line 143.

---

## Author Comment (AC3) · 10 May 2021

**Author reply to short comment from **Georgios Kazanidis** from 04 February 2021 (https://doi.org/10.5194/bg-2020-440-SC1) on:**

**Cushion bog plant community responses to passive warming in southern Patagonia**

Verónica Pancotto et al.

Reviewer comments (RC)
Author comments (AC)
Mentioned line numbers refer to the originally submitted manuscript
Manuscript changes (MC)

Comment on the under-review manuscript "Cushion bog plant community responses to passive warming in Southern Patagonia" by: Erika Hodgson*, Naomi Gunasekara*, Jonathan Garrido-Mirapeix Munn*, Ryan Newman*, Kai Westwell*, Georgios Kazanidis** *Undergraduate student in the course "Critical Thinking in Ecological and Environmental Sciences" at the University of Edinburgh **Tutor in in the course "Critical Thinking in Ecological and Environmental Sciences" at the University of Edinburgh

Dear authors,

as part of the undergraduate course "Critical Thinking in Ecological and Environmental Sciences" at the University of Edinburgh we have read carefully the mentioned above manuscript and we would like to express here our thoughts. We have found this piece of work is, in overall, a timely and interesting manuscript and we hope that our thoughts will help the authors to improve the status of their under-review paper.

**Abstract**

We feel that that the abstract is well written with a use of clear language. We think, however, that the authors should highlight/elaborate further on some of their interesting findings. E.g., we think that they should have highlighted that while some aspects of the plant's biology were affected by the rising temperatures (e.g., biometric features, photosynthesis, respiration) others (e.g., pigments) were not; suggestions explaining this "divergence" would be welcome.

The list of what was not affected by warming would be quite long; too long for the abstract in our opinion. The increase of pigments over time was indeed different between treatment and control plants as discussed in lines 238 – 241.

Also, we feel that the authors should have clarified which IPCC climate scenarios they have incorporated in their work.

Agreed, we changed the last sentence of the abstract to:

Our results suggest that even moderate future warming under the SSP1-2.6 scenario could decrease the carbon sink function of austral cushion bogs.

**Introduction**

The introduction has a good structure in overall. We think, however, that the authors could have addressed the particular research gap more thoroughly. For example, the authors should have explained why they have chosen to work with the species Astelia pumila e.g., is this a cosmopolitan species, does it have a key role in ecosystem structure

and functioning? Clarification on these aspects could increase the overall impact of the manuscript and its findings and make it more accessible.

Cushion bogs dominated by Astelia pumila are particularly understudied and are a common feature of the Magellanic Moorland, one of the largest peatland regions in the world. We do mention these motivations for our study.

We feel also that the "Introduction" would have benefited if the authors had made some null hypothesis about the impacts of the changing abiotic parameters on the species biology. We feel that the lasts parts of the Introduction should mention to the readers which are the main aims and objectives of the work and how the findings will fit into the larger picture. Currently the last parts of the Introduction (e.g., lines 41-45) should be removed to the "Materials and Methods" section of the paper.

We restructured the end of the introduction in response to the comments from both referees, largely addressing the above-mentioned points.

**Materials and Methods**

Lines 89-90: Please mention the measurements units used for measuring the size of the plant's leaves.

Unit (mm) added.

Line 96: Can you please explain how the number "86" has been reached/calculated?

See Table S2, 86 is the number of leaf samples we managed to collect.

The current number of replicates (n=3) for the semicircular plastic walls is acceptable; however higher numbers could provide higher statistical robustness. We acknowledge that logistical constraints may have prevented the use of these plastic walls.

We suspect a misunderstanding here coming from us mentioning n=3 in the abstract. There were 10 treatment plots of which three were equipped with temperature sensors. We changed the abstract to:

We installed a year-round passive warming experiment using semicircular plastic walls that raised average near-surface air temperatures between 0.4 C and 0.7 C (at the three of ten treatment plots which were equipped with temperature sensors).

There was also no mention of the number of individual organisms present per plot.

See Table B2 for plant coverage data.

Line 113: The sampling dates were mentioned, but it was not said how many replicate measurements were taken for the the $CO_2$ flux measurements on the treatment and control plots during this time. We do acknowledge that this information was in appendix A1.

As stated in responses to similar issues by both referees: In our sampling design and due to the high temporal variability of $CO_2$ fluxes, measuring fluxes over a wide range of light and temperature conditions was prioritized over measuring at control and treatment plots on the same day or equally often. We therefore did not exactly collect replicate measurements in that sense that we were aiming to e.g. compare averages, but we collected data to fit response functions to in order to compare model parameters.

**Results**

Lines 185-210 ("Treatments effects on temperature"). We think that these lines would fit better in the "Materials and Methods" section.

Verifying that the method achieved what it was supposed to is a result in our opinion.

Lines 215-218 suggest that the growing season ranges from September to April. Please clearly define the range of the growing season in the "Materials and Methods" section.

The southern hemisphere growing season does range from September to March, see Table B1.

Figure 3, panel d (September 2017). Based on the p-values that are provided the differences are not statistically significant (p>0.05). The authors state in the caption that ". . .this difference is less significant (p < 0.1)" which sounds a bit odd.
Changed to:
In September 2017, leaf lengths are only different at a lower significance level (p < 0.1)

Table 1. Please mention in the caption the Table the measurement units for the growth rate.
Done, "µm/d" added.

Figure 4. Some of the information provided in the caption is rather redundant (e.g. We divided the area estimates into two groups referring to midsummer and late summer (see supplementary Table S2) and compared the respective treatment and control means using a Mann-Whitneytest). We feel that this level of details is not necessary in here and would be adequate if it is only shown in the "Materials and Methods", section.
In our opinion, it is easier to understand what the p-value in the figure refers to if the method is briefly described in the caption.

**Discussion**

We feel that it is not the best way to start a Discussion by highlighting technical aspects; instead the authors could have given a succinct overview of the major/most interesting findings based on which they will build their Discussion. Studying the impacts of rising temperature on plant biology is a key feature; however, it is common knowledge that it is not the parameters that exerts stress on organisms; based on that it would be welcome if authors mention that a multiple-stressor experiment would have provided a better insight about the effects of climate change. Also, elaboration (even a succinct one) on the findings of other relevant studies about the impacts of multiple stressors on plant performance and implications about ecosystem structure and functioning (e.g., elemental cycling) would be great.
In our opinion, there are a few good reasons to be skeptical about the passive warming method we applied in our study. We think it is necessary to address those points, which might seem too technical for some readers, at such length.

**Conclusions**

The last part of the conclusions should have highlighted how the key findings of the present work fit into the bigger picture e.g. in the functioning and resilience of ecosystems where these plants are abundant. It may be beneficial to conclude by highlighting why your findings are relevant and potentially suggest management strategies to reduce the impact on Astelia pumila. It would be welcome also the authors to highlight some research gaps that would simulate future research works.
We agree that the conclusions were missing some key aspects. We rewrote the whole section.
We conducted a warming experiment in a southern hemisphere cushion bog to investigate responses of the cushion-forming plant Astelia pumila to elevated temperatures as they are projected to occur on the southern hemisphere in a future climate. At warmed plots, A. pumila grew in denser cushions and had shorter leaves leading to unchanged aboveground biomass per area.  Furthermore, A. pumila physiology was altered so that at warmed plots, photosynthesis was less efficient while respiration

was intensified. We propose an increase in photorespiration as a response to warming as one likely underlying mechanism since it could explain the diminished gross primary production and enhanced respiration simultaneously. Apart from alterations of the photosynthetic apparatus, differences in leaf morphology and chlorophyll contents between treatment and control plants most likely additionally, or even decisively, contributed to the observed GPP variability. Respiration variability could additionally have been impacted by changes in root respiration and stress-induced enhanced photooxidation.

Over the main growing season of two exemplary years, warmed A. pumila cushions cumulatively took up 55 % and 85 % less $CO_2$-C than the cushions of unaltered control plots. This change in net C uptake is considerable, especially when comparing the amount of artificial warming achieved in our experiment (annual average between 0.4 °C and 0.7 °C at the three of ten replicates which were equipped with temperature sensors) with temperature projections for the region from the Coupled Model Intercomparison Project Phase 6 (CMIP6). Estimates for contrasting Shared Socioeconomic Pathways (SSPs) show increases in mean annual 2 m air temperature of 1 °C (SSP1-2.6) and 2 °C (SSP5-8.5) from 2014 to 2100 (Wieners et al. 2019a, b). In conjunction with our findings, a considerable weakening of the long-term C sink strength of austral cushion bogs in a future climate seems likely. However, the temporal cover of flux measurements in our study was biased towards the growing season and more data from the shoulder seasons and winter, when temperatures are lower but photosynthesis of the evergreen A. pumila is ongoing, would be desirable and should be collected in future studies.